# Functional role of the type 1 pilus rod structure in mediating host-pathogen interactions

**Caitlin N Spaulding[1,2†], Henry Louis Schreiber IV[1,2†], Weili Zheng[3†], Karen W Dodson[1,2], Jennie E Hazen[1,2], Matt S Conover[1,2], Fengbin Wang[3], Pontus Svenmarker[4], Areli Luna-Rico[5], Olivera Francetic[5], Magnus Andersson[4], Scott Hultgren[1,2*], Edward H Egelman[3]**

[1]Center for Women's Infectious Disease Research, Washington University School of Medicine, St. Louis, United States; [2]Department of Molecular Microbiology, Washington University School of Medicine, St. Louis, United States; [3]Department of Biochemistry and Molecular Genetics, University of Virginia, Charlottesville, United States; [4]Department of Physics, Umeå University, Umeå, Sweden; [5]Department of Structural Biology and Chemistry, Institut Pasteur, Biochemistry of Macromolecular Interactions Unit, Paris, France

**Abstract** Uropathogenic *E. coli* (UPEC), which cause urinary tract infections (UTI), utilize type 1 pili, a chaperone usher pathway (CUP) pilus, to cause UTI and colonize the gut. The pilus rod, comprised of repeating FimA subunits, provides a structural scaffold for displaying the tip adhesin, FimH. We solved the 4.2 Å resolution structure of the type 1 pilus rod using cryo-electron microscopy. Residues forming the interactive surfaces that determine the mechanical properties of the rod were maintained by selection based on a global alignment of *fimA* sequences. We identified mutations that did not alter pilus production in vitro but reduced the force required to unwind the rod. UPEC expressing these mutant pili were significantly attenuated in bladder infection and intestinal colonization in mice. This study elucidates an unappreciated functional role for the molecular spring-like property of type 1 pilus rods in host-pathogen interactions and carries important implications for other pilus-mediated diseases.
DOI: https://doi.org/10.7554/eLife.31662.001

*For correspondence:
hultgren@wustl.edu

†These authors contributed equally to this work

## Introduction

To mediate colonization of host and/or environmental habitats, Gram-negative bacteria encode a highly conserved family of adhesive pili called chaperone-usher pathway (CUP) pili. Notably, CUP pili are critical virulence factors in a wide range of pathogenic bacteria, including *Escherichia*, *Klebsiella*, *Pseudomonas*, *Haemophilus*, *Salmonella* and *Yersiniae* genera (*Nuccio and Bäumler, 2007*). To date, 38 distinct CUP pilus types have been identified in *Escherichia* and *Shigella* genomes and plasmids, each of which is hypothesized to promote bacterial colonization of a distinct habitat (*Nuccio and Bäumler, 2007*; *Wurpel et al., 2013*). Interestingly, single *Escherichia coli* genomes carry up to 16 distinct CUP operons, suggesting that the retention of an assortment of CUP pilus types by a single strain may be necessary to accommodate the complex lifecycle of *E. coli* (*Wurpel et al., 2013*).

Arguably, the best-studied CUP pili are those encoded by uropathogenic *E. coli* (UPEC), which is the causative agent of the majority of urinary tract infections (UTIs). UTIs affect 150 million people annually worldwide and are associated with significant morbidity and economic impact (*Foxman, 2014*). UTI treatment failure is common, with ~25% of woman suffering from recurrent UTI

**eLife digest** *Escherichia coli*, or *E. coli* for short, is a type of bacteria commonly found in the guts of people and animals. Certain types of *E. coli* can cause urinary tract infections (UTIs): they travel from the digestive tract up to the bladder (and sometimes to the kidneys) where they provoke painful symptoms. To cause the infection, the bacteria must become solidly attached to the lining of the bladder; otherwise they will get flushed out whenever urine is expelled.

Pili are hair-like structures that cover a bacterium and allow it to attach to surfaces. *E. coli* has many different types of pili, but one seems particularly important in UTIs: type 1 pili. These pili are formed of subunits that assemble into a long coil-shaped rod, which is tipped by adhesive molecules that can stick to body surfaces. The current hypothesis is that the pili act as shock absorbers: when the bladder empties, the pili's coil-like structure can unwind into a flexible straight fiber. This would take some of the forces off the adhesive molecules that are attached to the bladder, and help the bacteria to remain in place when urine flows out. However, the exact structure of type 1 pili is still unclear, and the essential role of their coil-like shape unconfirmed.

Here, Spaulding, Schreiber, Zheng et al. use a microscopy method called cryo-EM to reveal the structure of the type 1 pili at near atomic-level, and identify the key units necessary for their coiling properties. The experiments show that pili with certain mutations in these units unwind much more easily when the bacteria carrying them are 'tugged on' with molecular tweezers. The bacteria with mutant pili are also less able to cause UTIs in mice. The coiling ability of the type 1 pili is therefore essential for *E. coli* to invade and colonize the bladder.

Every year, over 150 million people worldwide experience a UTI; for 25% of women, the infection regularly returns. Antibiotics usually treat the problem but bacteria are becoming resistant to these drugs. New treatments could be designed if scientists understand what roles pili play in the infection mechanisms.

DOI: https://doi.org/10.7554/eLife.31662.002

(rUTI), and is due, in part, to the increasing prevalence of drug-resistant UPEC strains (*Scholes et al., 2000*; *Zowawi et al., 2015*). UPEC that infect the urinary tract often originate from the host gastrointestinal tract. After being shed from the gut in the feces, UPEC can colonize peri-urethral or vaginal areas and subsequently ascend through the urethra to the bladder and/or kidneys, instigating UTI. In mice, type 1 pili, which promote binding to mannosylated proteins, play critical roles in both the gut and urinary tract. Recent work has revealed that type 1 pili help mediate UPEC intestinal colonization, thus promoting the establishment and/or maintenance of the UPEC reservoir in the gut that can eventually seed UTI (*Spaulding et al., 2017*). Upon entering the bladder, type 1 pili facilitate bacterial colonization and subsequent invasion into epithelial cells lining the bladder lumen (*Mulvey et al., 1998*; *Martinez et al., 2000*). Bladder invasion is a critical step in UPEC pathogenesis, allowing the bacteria to replicate in a niche protected from innate immune defense mechanisms, antibiotics, and expulsion during urination. UPEC that cannot invade the urothelium, like those lacking type 1 pili or its associated adhesin, FimH, are quickly cleared from the bladder, emphasizing the importance of type 1 pili mediated host-pathogen interactions on the fitness of UPEC during cystitis (*Wright et al., 2005*). After invading into a bladder cell, UPEC escape into the host cell cytoplasm and replicate to form biofilm-like intracellular bacterial communities (IBCs) (*Anderson et al., 2003*; *Justice et al., 2004*) comprised of ~$10^4$ cells (*Wright et al., 2007*). Mouse models of UTI have revealed that while some mice are capable of self-resolving acute UPEC infection, others progress to chronic cystitis, which is characterized by persistent high titer bacteriuria (>$10^4$ CFU/ml) and high bacterial bladder burdens (>$10^4$ CFU) two or more weeks after inoculation (*Hannan et al., 2010*). In the absence of antibiotic treatment, chronic cystitis can also be observed in women (*Mabeck, 1972*; *Ferry et al., 2004*).

*E. coli* strains, including UPEC, are grouped into distinct clades (e.g., clades A, B1, B2, D, and E) based on their genetic relatedness (*Tenaillon et al., 2010*). While UPEC strains tend to be genetically heterogeneous the majority of UPEC strains isolated from women with UTI in the USA reside in the B2 clade (*Schreiber et al., 2017*). While the types of CUP pilus operons encoded in *E. coli* genomes varies between different clades and individual strains, the vast majority of sequenced *E.*

*coli* strains, including nearly all sequenced UPEC clinical isolates, carry an intact copy of the type 1 pilus (*Wurpel et al., 2013*; *Schreiber et al., 2017*). Type 1 pili are encoded by the *fim* operon which, like the gene clusters encoding other CUP pili, encodes all the dedicated proteins necessary to assemble a mature pilus onto the bacterial surface, including: an outer-membrane pore-forming usher protein, a periplasmic chaperone protein, pilus subunits, and the tip adhesin protein. Most pilus tip adhesins, including the FimH adhesin, are made up of two domains, an N-terminal lectin domain, which is responsible for recognition and attachment to a ligand(s), and a C-terminal pilin domain, which connects the adhesin to the bulk of the pilus (*Jones et al., 1992*). Pilus subunits, including the pilin domain of the adhesin, are comprised of an incomplete immunoglobulin (Ig)-like fold, which lacks the C-terminal β-strand and require the action of the dedicated periplasmic chaperone for proper folding (*Sauer et al., 2002*). In a process known as donor strand complementation (DSC), a periplasmic chaperone templates subunit folding by transiently providing one of its β-strands (*Sauer et al., 1999*). Chaperone-subunit complexes are then delivered to the outer membrane usher, which catalyzes pilus assembly *via* a reaction known as donor strand exchange (DSE). In DSE, the strand donated to a nascent subunit by the chaperone is replaced by the N-terminal extension (Nte) of an incoming subunit (*Sauer et al., 2002*).

Following this pattern, the type 1 pilus usher (FimD) and chaperone (FimC) help type 1 pili assemble into a composite pilus structure consisting of a short tip fibrillum made up of the adhesin protein (FimH) and two minor subunits (FimG and FimF) that is joined to the pilus rod, a homopolymer of ~1000 FimA subunits. Once extruded to the extracellular surface, the type 1 pilus coils into a rigid right-handed helical structure that is capable of unwinding into a flexible linear fiber (*Abraham et al., 1992*; *Jones et al., 1995*; *Saulino et al., 2000*; *Aprikian et al., 2011*). This ability to transition between a coiled, helical rod and a linear fiber has been proposed to allow the type 1 pilus to act as a 'molecular spring' to maintain adherence in the face of fluid shear forces (*Zakrisson et al., 2012*). Specifically, we hypothesize that in the absence of urine voiding, the type 1 pilus rod is maintained in the coiled helical state that permits subsequent contact and invasion into bound epithelial cells. However, upon encountering the shear forces associated with micturition, the pilus extends to the linear form, absorbing the shear force and thus preventing the expulsion of the bacteria from the bladder.

Here, using high-resolution cryo-electron microscopy (cryo-EM), we solved the structure of the type 1 pilus rod. Residues involved in critical FimA-FimA interactions were identified that when mutated reduced the force required to unwind the helix, despite not altering the ability of the FimA protein to be incorporated into the pilus rod. When introduced into the chromosome of UTI89, a human UPEC clinical cystitis isolate, these point mutations dramatically reduced the ability of UTI89 to establish an intestinal reservoir and cause acute and chronic cystitis. In contrast, these point mutant strains did not result in large scale differences in levels of piliation or ability to agglutinate mannose-expressing guinea pig red blood cells in vitro. Taken together our results show that the identity of residues within the FimA rod, are critical for the type 1 pili mediated virulence of UPEC in the urinary tract and suggest that the helical pilus rod has an important functional role, beyond serving as a platform to present FimH, in promoting colonization in the gut and infection of the bladder.

## Results

### Determination of the type 1 pilus rod structure

To characterize FimA polymers that form the type 1 pilus rod, we solved the cryo-EM structure of native type 1 pili appearing in a preparation of recombinantly expressed Type IV pili (T4P) from the *E. coli* K12 strain BW25113. As shown in *Figure 1A*, three types of filaments could be separated by eye: T4P, flagellar filaments, and a third class that were thicker and more rigid than T4P but thinner than the flagellar filaments. Sequencing the *fimA* PCR product showed that the encoded amino acid sequence was identical to the FimA protein from BW25113 (GenBank AIN34588.1) and MG1655 (GenBank NP_418734.1). There was no possibility of cross-contamination of the FimA and T4P filament images, as each has a very different helical symmetry. Averaged power spectra from FimA filaments (*Figure 1—figure supplement 1*) showed no trace of the Type IV pilus spectrum (*Figure 1—figure supplement 1*).

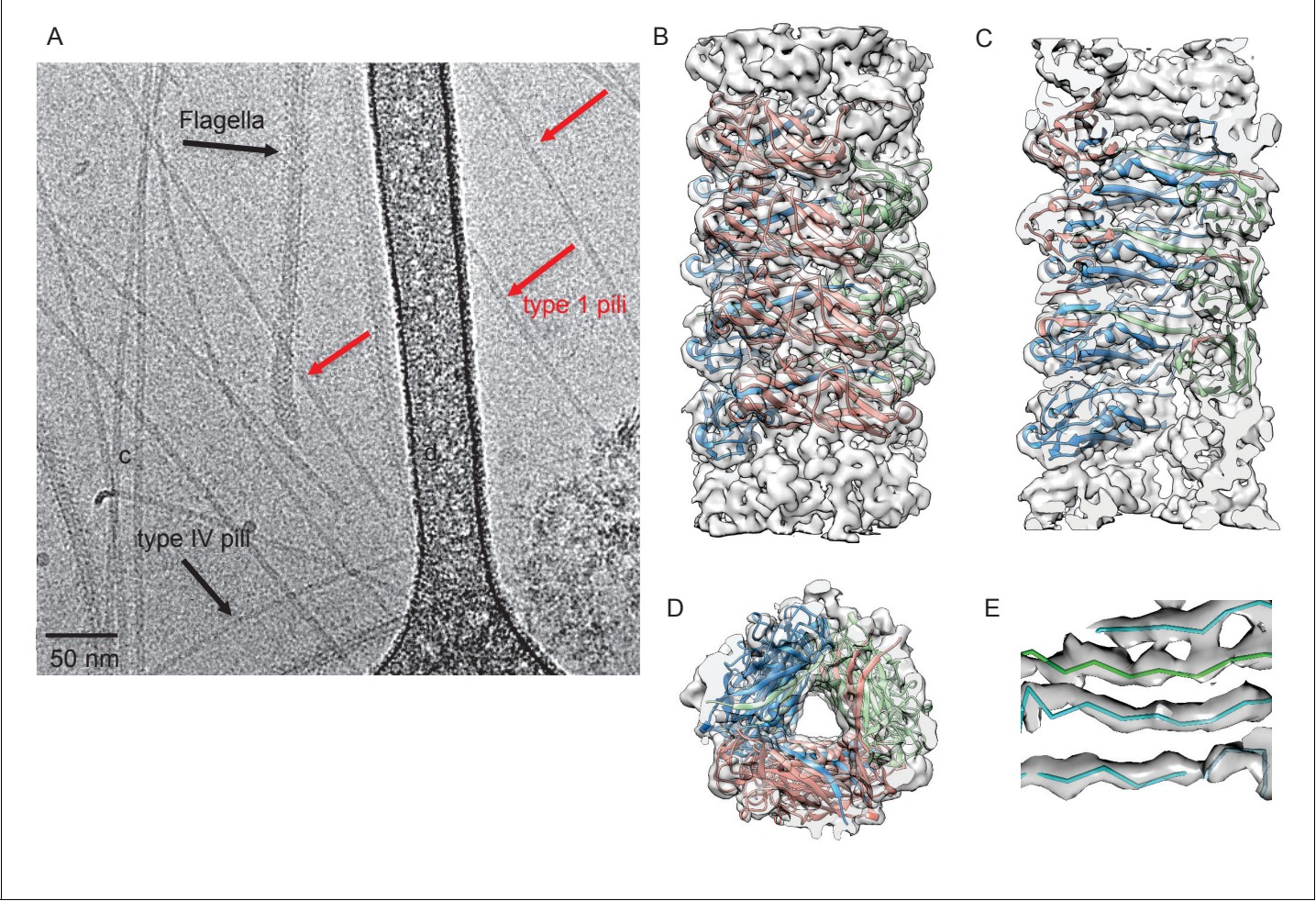

**Figure 1.** Cryo-EM Structure of Type 1 pili. (**A**) An electron micrograph of type 1 pili in vitreous ice, surrounded by type 4 pili and flagellar filaments. (**B**) Side view, (**C**) interior view (the front half of the reconstruction has been removed) and (**D**) top view of overall reconstruction of FimA rod with subunits along the same left-handed 3-start helix colored in either blue, salmon or green. (**E**) Close-up view of donor strand complementation (DSC) in the central lumen showing that the β-strands in the reconstruction are very well resolved.

DOI: https://doi.org/10.7554/eLife.31662.003

The following figure supplement is available for figure 1:

**Figure supplement 1.** Details of cryo-EM reconstruction of type 1 pili.

DOI: https://doi.org/10.7554/eLife.31662.004

The FimA reconstruction had an overall resolution of 4.2 Å (*Figure 1—figure supplement 1*), which is sufficient to build an atomic model of the structure (*DiMaio et al., 2015*). There were no ambiguities in threading the known FimA sequence through the density map (*Supplementary file 1*), even though the electron density observed on the outside of the rod is less well defined (*Figure 1B*). As a result, the first two residues and the last residue of FimA, which are located close to each other on the outside of the rod, are not resolved in the density map. The 4.2 Å estimate of the resolution is consistent with the well-separated β-strands and the density present for certain bulky side chains in the central lumen (*Figure 1C–E*). Further, in the structure we observe the N-terminal donor strand of one subunit completing the β-sheet of the adjacent subunit. Overall, the type 1 pilus is a 70 Å diameter rod where each adjacent subunit rotates around the helical axis by 115° and translates along the axis by 7.7 Å (*Figure 2A*). If we label each subunit along the 1-start helix by N, a subunit $N_0$ interacts with six adjacent subunits ($N_{-1}, N_{-2}, N_{-3}, N_{+1}, N_{+2}, N_{+3}$) (*Figure 2A,B*). The donor strand complementation involves mostly hydrophobic interactions. Most prominently, the hydrophobic amino acids Val5, Val10 and Phe12 in the N-terminal β-strand (Nte) of one subunit are

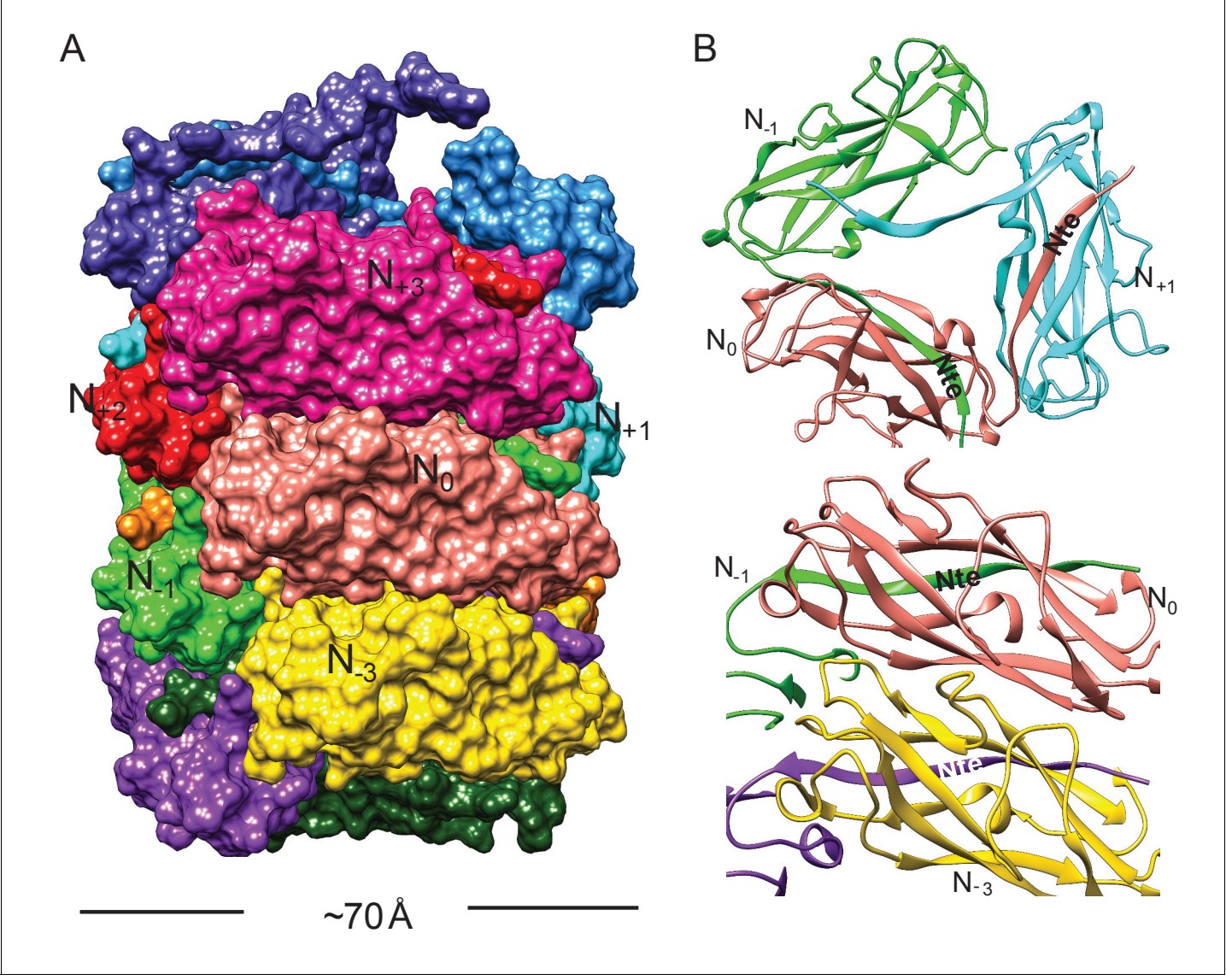

**Figure 2.** Subunit interface of the FimA rod. (**A**) Surface view of FimA rod model with subunits numbered along the right-handed 1-start helix. Each subunit is in a different color. (**B**) Ribbon representation to illustrate the interface of subunit $N_0$ (in salmon) with neighboring subunits.
DOI: https://doi.org/10.7554/eLife.31662.005

The following figure supplements are available for figure 2:

**Figure supplement 1.** Nte inserts into the hydrophobic groove of the neighboring subunit.
DOI: https://doi.org/10.7554/eLife.31662.006

**Figure supplement 2.** Comparison of subunit $N_0$ and $N_{+3}$ interfaces with previously deposited FimA pilus rod models.
DOI: https://doi.org/10.7554/eLife.31662.007

**Figure supplement 3.** Comparison of type 1 pili and P pili.
DOI: https://doi.org/10.7554/eLife.31662.008

inserted into the next subunit's hydrophobic groove created by a missing β-strand (*Figure 2—figure supplement 1*).

Further, residues corresponding to $FimA_{BW25113}$22–33, which are part of the interior surface of the rod helix, show structural differences between our cryo-EM rod structure and the previously solved crystal structures of FimA (*Crespo et al., 2012*)(*Figure 2—figure supplement 1*). In the DSC interaction of the FimA-FimC complex FimA residues 22–25 form a β-strand interaction with residues 59–56; and residues 31–33 form β-strand interactions with the FimC chaperone residues 102–104

(*Crespo et al., 2012*). Further, within a self-complemented DSE structure FimA residues 29–33 form a β-strand interaction with the appended Nte residues (9-13) (*Walczak et al., 2014*). Within the cryo-EM rod FimA structure, residues 25–29 adopt a helical conformation and residues 30–33 form a β-strand interaction with residues 10–13 from the Nte of the next FimA subunit. These residues form the center strand within the hollow helical core, and thus the ability of this region to adopt multiple conformations may be necessary in order for FimA to function as the major rod subunit.

## Comparison between P and type 1 pili

The interface between subunits $N_0$ and $N_{+3}$ is extremely important in maintaining the FimA helical rod. Previously, two atomic models (PDB ID: 2N2H and 2MX3) for the type 1 pilus have been generated using the same solid-state NMR data (*Habenstein et al., 2015*), with the helical symmetries for these models shown in *Supplementary file 2*. The results were described as being FimA from uropathogenic *E. coli* (UPEC) (*Habenstein et al., 2015*), however, the sequence corresponds to that of *E. coli* K12 strains, as does the recent FimA cryo-EM structure (*Hospenthal et al., 2017*) that is also described as being from UPEC. When subunit $N_0$ is superimposed between our cryo-EM model with a subunit in 2N2H and 2MX3, significant differences can be seen in the interface with subunit $N_{+3}$ in the different models (*Figure 2—figure supplement 2*). The overall RMSD between $C_\alpha$ atoms of subunit $N_{+3}$ in our cryo-EM model and 2N7H is 5.7 Å (*Figure 2—figure supplement 2*), while it is 11.0 Å with 2MX3 (*Figure 2—figure supplement 2*). Between 2MX3 and 2N2H, the overall RMSD is 5.6 Å for subunit $N_{+3}$ when subunits $N_0$ are aligned (*Figure 2—figure supplement 2*).

Comparison of our type 1 pilus model with the P pilus structure (*Hospenthal et al., 2016*) shows an interesting difference: in the P pilus, residues 1–5 are fully ordered and make a 90° bend to form a 'staple' which involves contacts with two other subunits (*Figure 2—figure supplement 3*). In contrast, the first two residues are disordered in our type 1 pilus rod, and the remaining three residues in this region project straight out of the structure and make no contacts with other subunits (*Figure 2—figure supplement 3*).

## Conservation and variability of FimA sequences

To explore the variation and evolution of FimA, we examined a set of 1,872 FimA protein sequences in the Ensembl Bacteria database (*Kersey et al., 2016*) (*Supplementary file 3*). After filtering to remove protein sequences that are expected to be non-functional and trimming of the signal sequence, the remaining 1,828 protein sequences were aligned. Here, we found that many of the FimA sequences were comprised of 159 residues, including BW25113, while other FimA sequences, such as the one from UTI89, a prototypical UPEC isolate, contained an additional two amino acid residues at the N-terminus of the mature protein (*Figure 3A*). Global alignment revealed a consensus sequence of 161 residues where half of the FimA residues in the mature protein are invariant (79/161 residues) and that 83.9% are highly conserved (139/161 residues with >95% sequence identity) (*Figure 3A*). We found the FimA sequences to be significantly more variable than other parts of the type 1 pilus machinery. For example, the mature forms of the FimC chaperone and FimH adhesin (n = 1,760 and n = 1,943, respectively; *Supplementary file 4*) obtained from the same database contain 97.6% and 97.5% highly conserved residues, respectively, as compared to 83.9% in FimA. In general, most *E. coli* genes evolve slowly (*Lee et al., 2012*), so the degree of divergence between alleles of FimA is abnormally high. This suggests that the selection forces acting on the FimA protein are different from the forces acting on the rest of the *fim* operon. Interestingly, the non-conserved residues within the FimA sequences are all located on the exterior surface of the solved cryo-EM model (*Figure 3B*), which is likely indicative of immune pressures selecting for antigenic diversification (*Wildschutte et al., 2004*). In particular, the first few amino acids of the N-terminus of the mature protein, disordered in our reconstruction, have some of the highest variability throughout the entire protein, suggesting that these amino acids do not play a functional role in the dynamics of unwinding in the type one pilus rod. In contrast, almost every residue involved in subunit-subunit interactions in the rod is either highly conserved or invariant (*Figure 3B,C*). This includes every residue lining the lumen (*Figure 3C*). The conservation of residues lining the lumen might be due to the transport of some product through this opening that is ~15 Å across (the height of the triangular lumen that rotates as one travels along the pilus). More likely, however, every residue in the lumen is

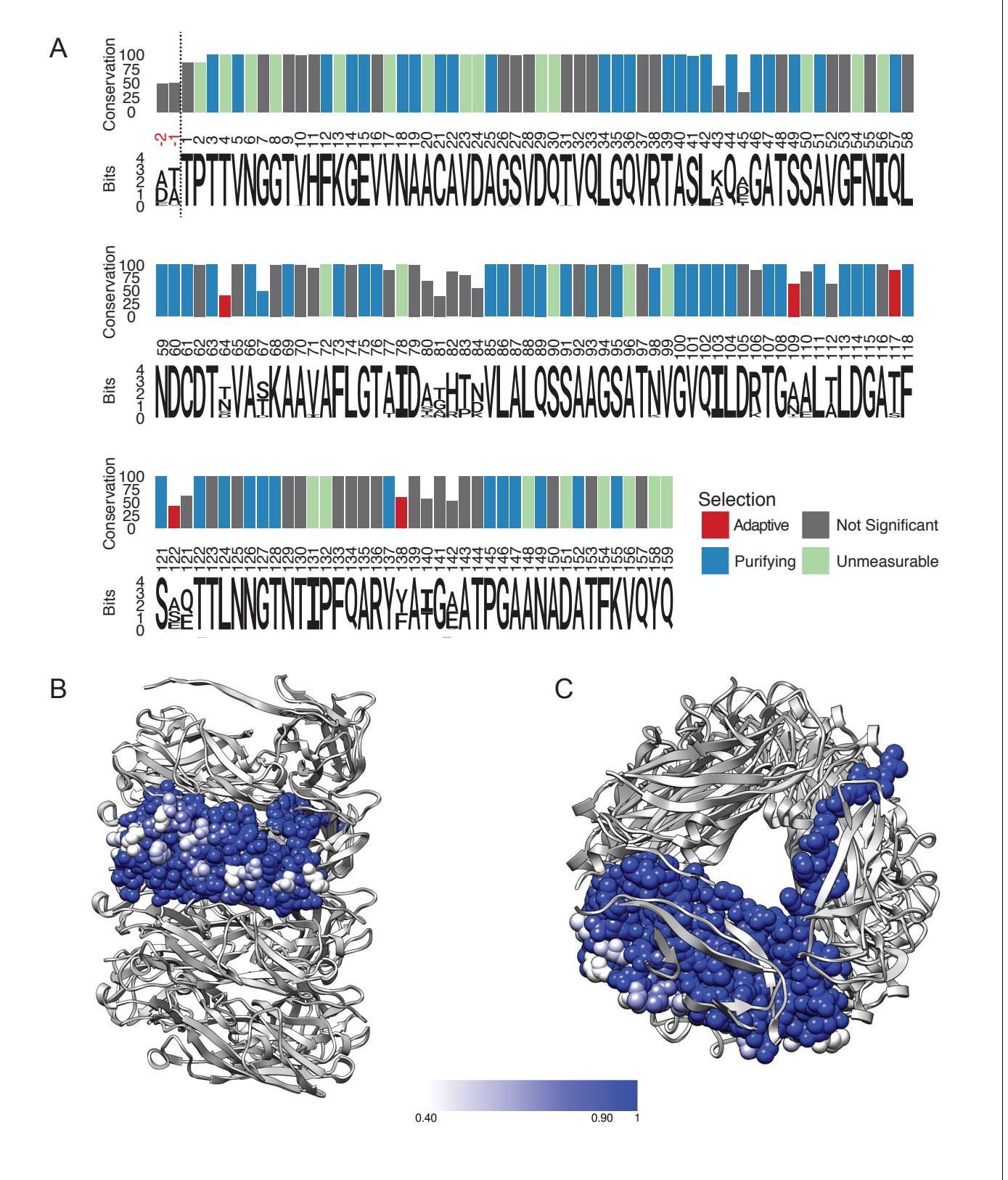

**Figure 3.** FimA conservation and variability. (**A**) Conservation, consensus amino acid identity, and selection pressures on residues within the mature form of the FimA protein measured by alignment of 1,828 sequences. Numbering here is based on FimA from BW25113 and the vertical dashed line indicates the location of the two amino acid residues, in red, that are present in some FimA sequences. Consensus amino acid residues are shown at
*Figure 3 continued on next page*

*Figure 3 continued*

each position. Height of the residues corresponds to their information content (bits) where the larger size indicates greater certainty that residue shown is the consensus residue at that position. The vertical bars represent the proportion of strains with the consensus amino acid (0–100%). Vertical bars are colored by selection pressure acting upon the amino acid position, with red and blue bars indicating residues with codons under adaptive and purifying selection, respectively, as measured by a $\chi^2$ (chi-squared) test. Green bars indicate codons with too little variability for evolutionary analysis and grey indicates amino acid positions that do not show statistically significant evidence for selection. (B,C) The degree of conservation for every FimA residue shown in (A) has been mapped onto a single subunit in our rod model. The absolutely conserved residues are in blue, the residues that are 40% conserved (the greatest degree of variability found) are in white, and light blue represents 90% conservation. Every residue facing the lumen is 100% conserved. Overall, FimA displays strong signals of purifying selection resulting in conservation with hotspots of adaptive selection resulting in variability of surface residues.

DOI: https://doi.org/10.7554/eLife.31662.009

The following figure supplement is available for figure 3:

**Figure supplement 1.** Location of FimA residues under positive selection for change in the helical rod.
DOI: https://doi.org/10.7554/eLife.31662.010

either involved in making a subunit-subunit contact (including the DSE contacts with the Nte from an adjacent subunit) or packed tightly between residues that are making such contacts.

To examine the importance of FimA-FimA interactions on bacterial pathogenesis, we chose to investigate how mutations in the *fimA* sequence of BW25113 altered functional and mechanical characteristics of type one pili. In addition, to assess whether FimA mutants altered the type-1 mediated ability of UPEC to infect the bladder and colonize the gut, we made mutations in the chromosomal *fimA* gene of UTI89. Thus, we do note that FimA_{UTI89} differs from our cryo-EM rod FimA_{BW25113} sequence in 16 residues, with all of the differences lying within highly variable residues located on the exterior of the pilus rod, including two additional amino acids at the N-terminus. These two additional N-terminal residues in UTI89 generate a shift in the numbering of UTI89 residues compared to FimA_{BW25113} (e.g., A22R in BW25113 corresponds to A24R in UTI89). To avoid confusion herein all residues are discussed using the BW25113 numbering system.

## Evidence of evolutionary pressures on FimA

To evaluate the evolutionary pressures shaping FimA and thus the type one pilus rod, we obtained gene sequences encoding the 1,828 FimA proteins from the Ensemble database to measure the selection pressures acting on the gene. After removing incomplete and duplicate sequences from the analysis, we were able to compare a total of 191 unique *fimA* sequences (*Supplementary file 3*), which were then trimmed to remove the signal sequence. These sequences encoding the mature protein were examined for evidence of recombination, which can result from horizontal transfer of genetic sequences between distinct strains and confound evolutionary analyses that assume all sequences were vertically inherited (*Anisimova et al., 2003*). Phylogenetic trees were thus corrected for use in subsequent analyses (*Kosakovsky Pond et al., 2006*). We then assessed the ratio of the rates of nonsynonymous (*dN*) to synonymous nucleotide mutations (*dS*) in each codon in the alignment (*dN/dS*) to estimate the selection pressures acting on each residue in the FimA protein using a fixed-effects likelihood measurement algorithm (*Kosakovsky Pond and Frost, 2005*). In general, a *dN/dS* ratio >1 is indicative of selection pressure favoring change in amino acid identity at the position (i.e., adaptive or positive selection) while a *dN/dS* ratio <1 indicates conservation at the codon (i.e., purifying selection or negative selection). Here, we found that 66 codons were under purifying selection, five codons were under adaptive selection, and an additional 27 codons were too conserved to be included in analysis (*Supplementary file 5*). The remaining 63 codons showed no evidence of statistically significant selection in either direction. All of the positively selected sites, N64, A109, T117, S120 and F138, encode residues located on the exterior surface of the FimA rod and away from any subunit-subunit interface (*Figure 3—figure supplement 1*). The 66 codons under purifying selection combined with the 27 codons too conserved for analysis encode many of the highly conserved residues found throughout subunit-subunit interfaces with the subunits above, below, and alongside (*Figure 3B,C*). This strongly suggests that evolutionary pressures are working to conserve the identity of residues that interact within the helical rod, while diversifying amino acid residues that are exposed to the host and susceptible to immune recognition.

To more specifically determine if variation in FimA correlated with *E. coli* clades or pathogenic lifestyles, such as uropathogenicity, we examined the diversity and distribution of the *fimA* gene in a curated set of 67 *E. coli* genomes. This dataset included 21 distinct UPEC strains isolated from a cohort of women with frequent, recurrent UTI and 46 reference *E. coli* strains that included lab and commensal strains, as well as a variety of intestinal and extra-intestinal pathogens (*Schreiber et al., 2017*)(*Supplementary file 6*). The *fimA* gene was carried by the majority of *E. coli* strains analyzed (57/67); including nearly every UPEC strain (96.3% or 26/27 strains). Importantly, the *fimA* sequences from UPEC strains were spread through the phylogenetic tree, indicating that there was not a single variant of FimA found in all UPEC strains (*Figure 3—figure supplement 1*). To determine if these different variants were under different selection pressure, we measured the *dN/dS* ratio of each branch in this *fimA* phylogeny to see if there were branches that were under different selection pressures than the rest of the branches in the phylogeny (i.e., the 'background' rate of selection) (*Kosakovsky Pond et al., 2011*). Here, we identified three branches with statistically significant evidence of episodic diversifying selection, including two branches carrying most of the clinical UPEC strains (branches labeled A and B) as well as enterohemorrhagic and enterotoxigenic *E. coli* strains in the branch labeled C (*Figure 3—figure supplement 1*). This pattern of evolution is indicative of strong adaptive selection acting on some, but not all, branches in a phylogeny. Taken together, we find that *fimA* is much more diverse than other parts of the type 1 machinery, owing to high rates of nucleotide polymorphisms and genetic recombination, and that *fimA* has undergone repeated rounds of strong selective pressures that have conserved residues responsible for subunit-subunit interfaces while varying the external surface of the protein that is exposed to the host milieu.

## Analysis of the effect of FimA mutations on Pilus expression and function in vitro

To determine which FimA residues are required to form the helical rod, we constructed single amino acid codon mutations in the BW25113 *fimA* gene, which were subsequently cloned into the expression vector, pTRC99a (*Amann et al., 1988*). Residues were changed to either Arginine or Glutamine to insert large charged residues that would promote disruption of FimA interactions without making the surface more hydrophobic. We expressed these variant *fimA* genes in trans in UTI89-LONΔ*fimA*, a strain with a chromosomal deletion of the *fimA* gene in a UTI89 strain where the phase-variable *fimS* promoter is locked in the ON orientation (LON) by altering the left inverted repeat necessary for promoter inversion (*Kostakioti et al., 2012*). This strain transcribes the *fim* operon constitutively, which removes the possibility that differences in type 1 pili assembly/function are due to effects on phase variation by the mutants.

We assessed the function of the type 1 pilus in each *fimA* mutant by measuring the agglutination of guinea pig red blood cells (GP-RBCs). Most of our *fimA* mutants showed hemagglutination (HA) titers similar to the wild-type (WT) *fimA* (*Supplementary file 7*). However, several mutations (A25R, V32R, V65R, V85R, and P145R) abolished the production of adhesive pili (*Supplementary file 7*). This suggests that these mutations, which are located in areas of high conservation throughout the FimA monomer, may disrupt critical interactions with FimC or FimD or may not be able to correctly fold, thus preventing the assembly of the pilus. Notably, each of these residues was either under strong purifying selection pressure (A25, V85 and P145) or was too conserved for evolutionary analysis (codons for V32 and V65 displayed no synonymous mutations and extremely limited non-synonymous mutation rates), which emphasizes their importance in assembly of the type 1 pilus rod (*Figure 3A*). Expression of some mutant *fimA* genes, particularly V5R, E45R, E121R and A142R, resulted in a bacterial clumping phenotype when grown in static culture. D62R and D114R also instigated bacterial clumping, but to a lesser extent. Three of these substitutions, E45R, E121R and A142R, are found in residues with relatively high variability in our analysis of FimA sequences, with just 33.9%, 62.3%, and 52.1% sequence identity, respectively. Further, we found that two of these residues (V5 and D114) showed statistically significant evidence of strong purifying selection (*Figure 3A*, *Supplementary file 6*).

To measure the impact of FimA$_{BW25113}$ variants produced by the UTI89-LONΔ*fimA* strain on pili force-extension responses, we chose six FimA mutant strains that did not disrupt pilus formation or produce severe bacterial clumping in vitro (A22R, A92R, D114R, D62R, K155R and P132R) and applied optical tweezers (*Figure 4—figure supplement 1*). Since type 1 pili are assembled from FimA subunits into a helical coil, they normally extend in three force phases: linearly increasing,

constant force, linearly increasing – where the characteristic constant force originates from sequential unwinding of subunits (region between the dashed lines in *Figure 4A*). The characteristic constant force extension enables the clear identification of instances where multiple pili are attached to a single microsphere, thus enabling us to determine the unwinding force of a single pilus (*Figure 4— figure supplement 2*). The unwinding force directly relates to the strength of layer-to-layer interactions (e.g., between subunits $N_0$ and $N_{+3}$), thus any changes in these interactions caused by mutations would shift the magnitude of the force plateau. To investigate this, we first measured unwinding force of single pili in the strain expressing WT FimA and subsequently examined the force response for each *fimA* mutant. The unwinding force of WT pili was 30.3 ± 0.2 pN (mean ±standard error (SE)) whereas all mutants showed a reduction in the force required for pilus unwinding (*Figure 4B*). The unwinding force measured for WT type 1 pili in this study is similar to what has been reported in the literature (*Andersson et al., 2007*). In particular, the A22R variant showed the largest reduction in the unwinding force, 11.2 ± 0.2 pN. We can explain this by the fact that Ala22 in our rod model is tightly packed against Ala93 from another subunit and Val37 from a third subunit (*Figure 4C*). Thus, replacing a small alanine with a long arginine side chain is expected to disrupt this interface.

## FimA mutants alter UPEC pathogenesis in the bladder and colonization of the gut

In the bladder, type 1 pili are required for binding and invasion of UPEC into superficial facet cells that line the bladder lumen and for the formation of intracellular bacterial communities (IBCs) during

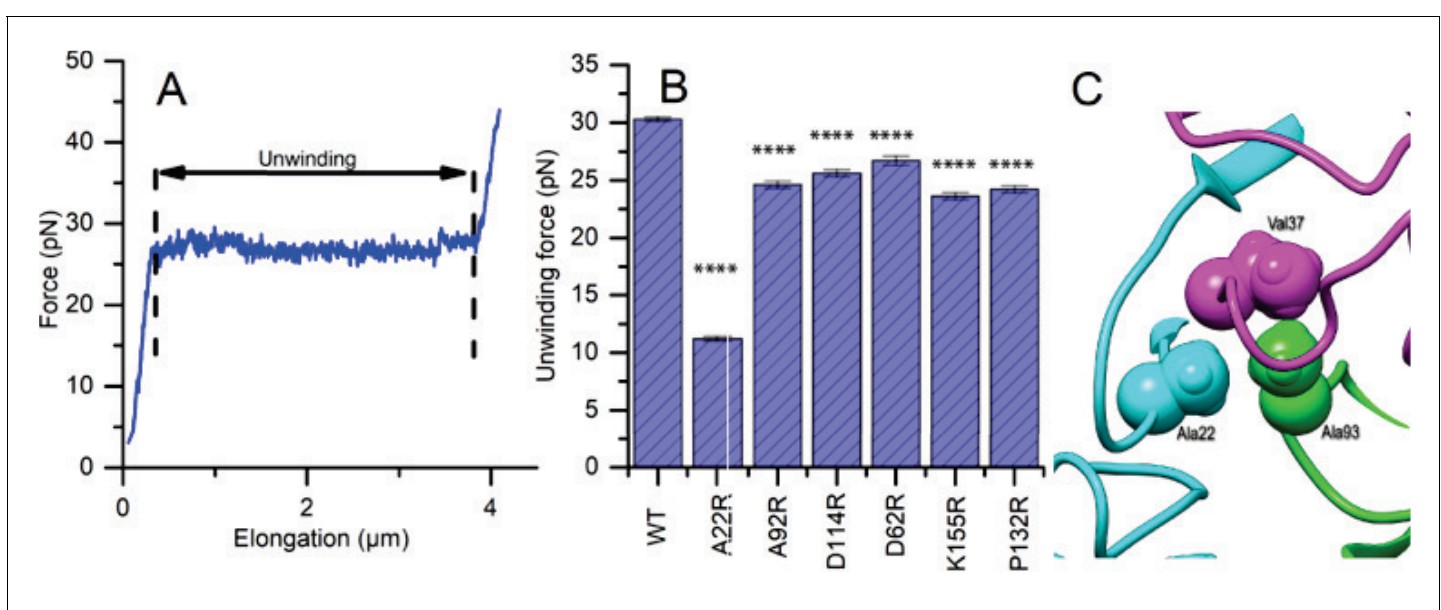

**Figure 4.** Mutations to FimA alter the force required to unwind the pilus. (**A**) Force response of a single type 1 pilus. The force response is composed of three phases, elastic stretching of the shaft, unwinding of the shaft, and elastic stretching of individual subunits in an open coil. (**B**) Bar chart showing the average unwinding force of FimA_BW254113 wildtype (WT) and mutants expressed in trans in UTI89-LONΔ*fimA* (**C**) Ala22 (cyan) plays an important role in the inter-subunit contacts through interacting with Val37 (magenta) from a second subunit and Ala93 (green) from a third subunit. (**B**) Bar chart showing the average unwinding force of measured pili where the error bars represent the standard error of the mean [WT, 30.3 ± 0.2 pN, N = 13, 2 replicates; A22R, 11.2 ± 0.2 pN, N = 25, 2 replicates; A92R, 24.6 ± 0.3 pN, N = 19, 2 replicates; D114R, 25.6 ± 0.3 pN, N = 16, 2 replicates; D62R, 26.7 ± 0.4 pN, N = 16, 2 replicates; K155R, 23.6 ± 0.3 pN, N = 14, 2 replicates; P132R, 24.1 ± 0.3 pN, N = 15, 2 replicates]. All replicates biological. ****p<0.0001 by unpaired, two-sided t-test).

DOI: https://doi.org/10.7554/eLife.31662.011

The following figure supplements are available for figure 4:

**Figure supplement 1.** Schematic illustration of the optical tweezer setup.
DOI: https://doi.org/10.7554/eLife.31662.012

**Figure supplement 2.** Force spectroscopy measurement showing attachment of multiple pili (A22R).
DOI: https://doi.org/10.7554/eLife.31662.013

the first 6–18 hr of acute infection. This has been shown to be critical for ongoing infection in both humans and mouse models of cystitis (*Spaulding and Hultgren, 2016*). To determine if mutations in the FimA sequence altered UPEC pathogenesis in relevant mouse models, we constructed UTI89 strains with clean single codon mutations in the chromosomal *fimA* gene (with the phase-variable *fimS* promoter intact). For these studies, we chose to investigate the effects of four FimA mutants that were examined via optical tweezers. Each of these mutations were made in the codon of a highly conserved amino acid positions of FimA$_{UTI89}$ (A22, D62, D114, and P132).

When compared to a strain with the reintegrated WT UTI89 *fimA* sequence, we found that all reintegrated FimA mutant strains demonstrated similar levels of GP-RBC agglutination, in vitro (*Figure 5—figure supplement 1*), despite some variability in the level of piliation between FimA mutant strain (*Figure 5—figure supplement 1*). Interestingly, those mutants that show some increased hemagglutination in the presence of exogenous mannose compared to the WT strain correlated with those that showed some clumping in culture. Overall, these findings indicate that the FimA mutations, which reduced the unwinding force of the rod, do not prevent the expression or function of type 1 pili in vitro.

We next investigated how each mutant altered the kinetics of bladder infection during competitive infections with WT UTI89 over 28 days. In mice that developed chronic cystitis (defined as the development of persistent high titer (>10$^4$ cfu/ml) bacteriuria and high titer (>10$^4$ cfu/ml) bladder bacterial burdens at sacrifice >4 weeks post-infection), strains producing FimA$_{UTI89}$ variants with the D62R, D114R, and A22R substitutions were outcompeted by up to six logs by the reintegrated WT strain (*Figure 5A–E*). The FimA P132R variant had no effect on the ability of the strain to compete with the WT strain in chronically infected mice (*Figure 5B*). Mice that resolved their infections (defined as any animal whose urine or bladder titers dropped below 10$^4$ cfu/ml at least once during the 4 week infection) in this experiment are shown in *Figure 5—figure supplement 2*.

In mice infected with a single UTI89 strain, the WT and P132R variant caused chronic cystitis at similar rates (45% and 30%, respectively) (*Figure 5F*, *Figure 5—figure supplement 3*). However, the D114R and D62R mutant strains had reduced rates of chronic cystitis of 20% and 10%, respectively. Interestingly, 100% of mice infected with the A22R variant resolved their infections over the 4 week experiment, with half of the mice (10/20) exhibiting sterile urines by 10 dpi compared to just 15% (3/20) of mice infected with the WT strain at the same time point (*Figure 5—figure supplement 3*). The defect in chronic infection caused by *fimA* variants is likely due to pathogenic deficiencies during acute UTI. Accordingly, we found that two *fimA* variants (A22R and D114R) significantly altered the ability of UTI89 to form IBCs at 6 hr post infection (hpi) (*Figure 5G*), with 8/10 mice infected with the D114R variant forming <15 IBCs. Even more strikingly, 70% of mice infected with the A22R variant formed no IBCs at 6 hpi and the other 30% formed three or less. This is in stark contrast to WT and P132R strains, which formed between 50–100 IBCs on average by 6hpi. Interestingly, the A22R mutant was severely attenuated in its ability to invade into bladder cells starting as early as 1 hr post infection (*Figure 5H*). The D62R variant was defective in chronic cystitis in both the competitive and single bladder infections, but no acute fitness defects were observed. Further, mice infected with D62R formed a similar number of IBCs as the WT strain at 6 hpi, suggesting that the defect occurs at a later time-point. Accordingly, in mice singly infected with *fimA* D62R bacterial clearance is delayed, starting between 10–14 days (*Figure 5—figure supplement 3*).

In addition to playing a pivotal role in the urinary tract, a recent study found that type 1 pili also promote the establishment and/or maintenance of the UPEC intestinal reservoir (*Spaulding et al., 2017*). Deletion of the operon encoding the type 1 pilus or the FimH adhesin impedes intestinal colonization by UTI89 (*Spaulding et al., 2017*). Therefore, we tested the impact of each of our four chromosomal FimA genetic variants on UPEC intestinal colonization levels. We found that strains producing FimA$_{UTI89}$ variants D62R, D114R, and A22R showed lower levels of intestinal colonization (by up to 2 logs) in the feces, cecum, and colon of mice compared to those colonized by WT *fimA* (*Figure 6*). This 2-log decrease mirrors the defect observed in UTI89Δ*fim* or UTI89Δ*fimH* strains, suggesting that these mutations prevent type 1 pilus-dependent gut colonization in UTI89 (*Spaulding et al., 2017*). While not statistically significant, *fimA*$_{UTI89}$ P132R variant showed lower colonization (by up to one log) in the feces and cecum than the WT strain (*Figure 6*). Together, our data indicates that the sequence of the FimA major subunit is critical for pilus function in the bladder and gut and thus has a major impact on the outcome of UTI.

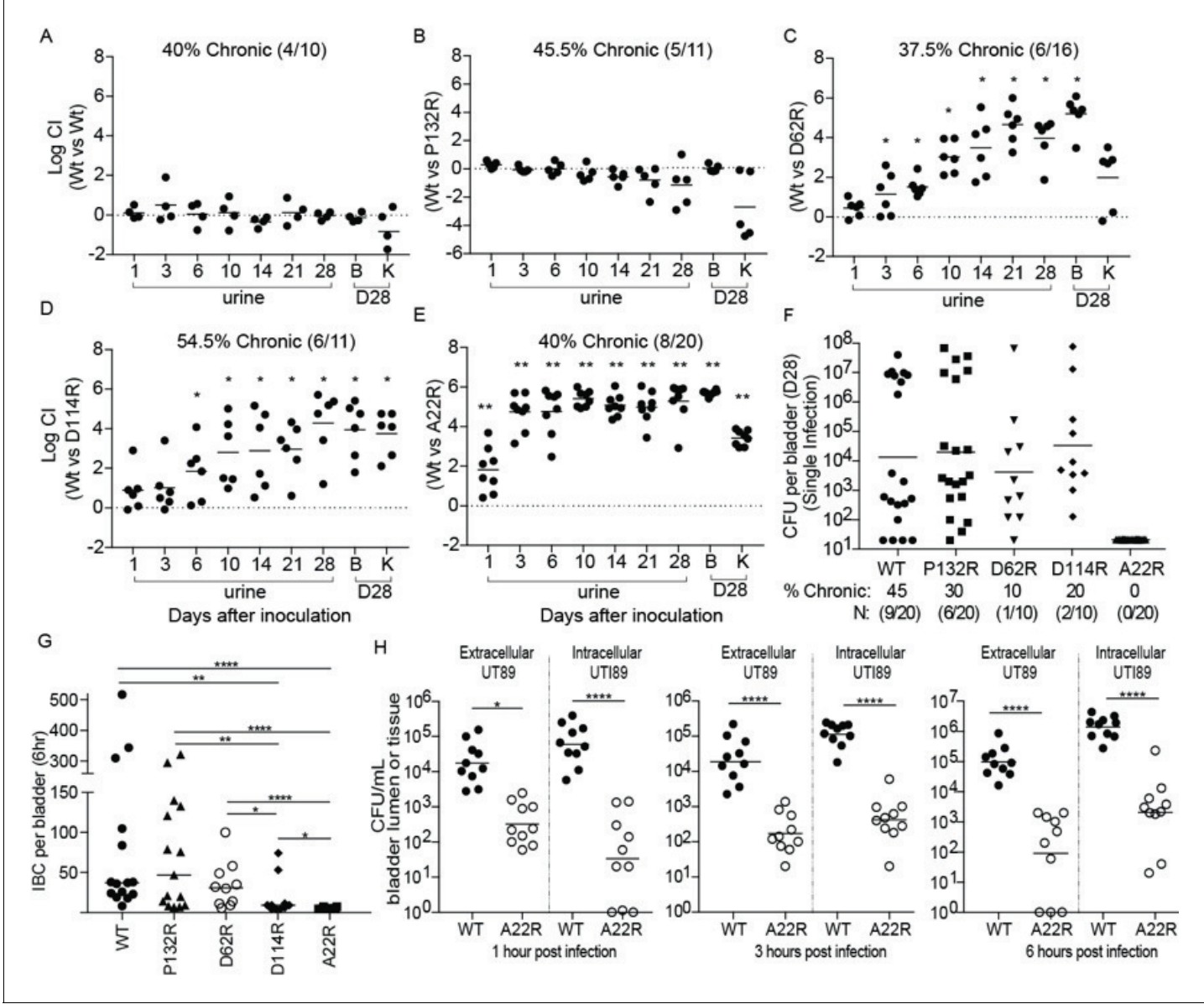

**Figure 5.** Point mutations in FimA alter UPEC pathogenesis in the bladder. C3H/HeN mice were concurrently transurethrally inoculated with $1 \times 10^8$ CFU of wildtype (WT) UTI89 and one of four isogenic UTI89 strains containing point mutations in the *fimA* gene. (A–E) Longitudinal urinalyses were performed over 28 days and examination of UTI89 and FimA mutant titers in bladders and kidneys were determined at time of sacrifice (28 dpi). (F) Mice were also infected singly, via transurethral inoculation, with UTI89 or FimA mutant strains. Longitudinal urinalysis was perfomed over 28 days (*Figure 5—figure supplement 3*). Bladders titers taken at 28 dpi after single infections are shown in panel F along with the percentage of mice that developed chronic UTI (% chronic) along with the number of mice used in the study (N). (G) Number of intracellular bacterial communities (IBCs) formed by indicated strain at 6 hr post infection (hpi). (H) Mice were singly infected with WT or A22R mutant and the ability of each strain to invade into bladder tissue was assed as 1, 3, and 6 hr post infection. Intracellular bacteria were able to invade into the bladder tissue while extracellular bacteria were detected in the bladder lumen. Abbreviations. B = bladder, K = kidney, CI = competitive index. Bars represent mean (A–E), geometric mean (F, H), and median (G). *p<0.05, **p<0.01, ***p<0.001, ****p<0.0001 by Wilcoxon Signed Ranked test (A–F) or Mann Whitney U test (G, H). N = 10, 2 replicates (A, G, H); N = 11, 2 replicates (B, E); N = 16, 3 replictates (C); N = 18, 3 replicates (D); N = 20 mice, 4 replicates (F, G); N = 15, 3 replicates, (H); N = 4, 2 replicates (I). All replicates are biological.

DOI: https://doi.org/10.7554/eLife.31662.014

The following figure supplements are available for figure 5:

**Figure supplement 1.** Chromosomally integrated point mutations in *fimA* gene do not prevent the expression or function of type 1 pili in vitro.
DOI: https://doi.org/10.7554/eLife.31662.015

**Figure supplement 2.** CFU titers for mice that resolved competitive bladder infections shown in *Figure 5*.
*Figure 5 continued on next page*

*Figure 5 continued*

DOI: https://doi.org/10.7554/eLife.31662.016

**Figure supplement 3.** CFU titers for mice that developed chronic UTI or resolved infection in single bladder infections shown in Figure 5.

DOI: https://doi.org/10.7554/eLife.31662.017

## Discussion

### The power of Cryo-EM to define the structure of helical polymers

Tremendous advances in cryo-EM within the past four years (*Egelman, 2016*; *Subramaniam et al., 2016*) largely driven by the introduction of direct electron detectors (*Li et al., 2013*) has meant that many complexes that were recalcitrant to crystallization can now be readily solved at near-atomic resolution by cryo-EM. In particular, it is exceedingly difficult to crystallize most helical polymers, as unless such a polymer has exactly two, three, four or six subunits per turn, it cannot be packed in any crystal space group so that all subunits are in equivalent environments. The type 1 pilin, FimA, has been extensively studied by x-ray crystallography and solution NMR, while the type 1 pilus has only been studied at high resolution by solid-state NMR (*Habenstein et al., 2015*). We show here that type 1 filaments, present as a background in a preparation of T4P, allow us to reach a near-atomic level of resolution and build an atomic model for the FimA rod. The ability of cryo-EM to separate out multiple conformations among particles (*Gui et al., 2017*; *Vonck and Mills, 2017*) or even of subunits within the same particle (*Roh et al., 2017*) has been one of the greatest strengths of cryo-EM, allowing for multiple states to be solved from the same micrographs. Biochemically heterogeneous preparations, such as when virions are a mixture of empty particles and those containing DNA (*Dong et al., 2017*) are now routinely sorted out into homogeneous structural classes. We have taken advantage of that strength here to show that filament images which might otherwise have been discarded as a background contaminant can actually be used to build an atomic model. Our type 1 pilus rod model provides new insights into structure-function relationships in type 1 pili.

### Residues responsible for the structure of FimA influences UPEC colonization of the bladder and gut

FimA is critical for proper assembly of the type 1 pilus and, as the major subunit that makes up the pilus rod, is also critical for the proper display of the FimH adhesin. However, here we have uncovered a more complex and previously unappreciated role for FimA and the type 1 pilus rod in host-

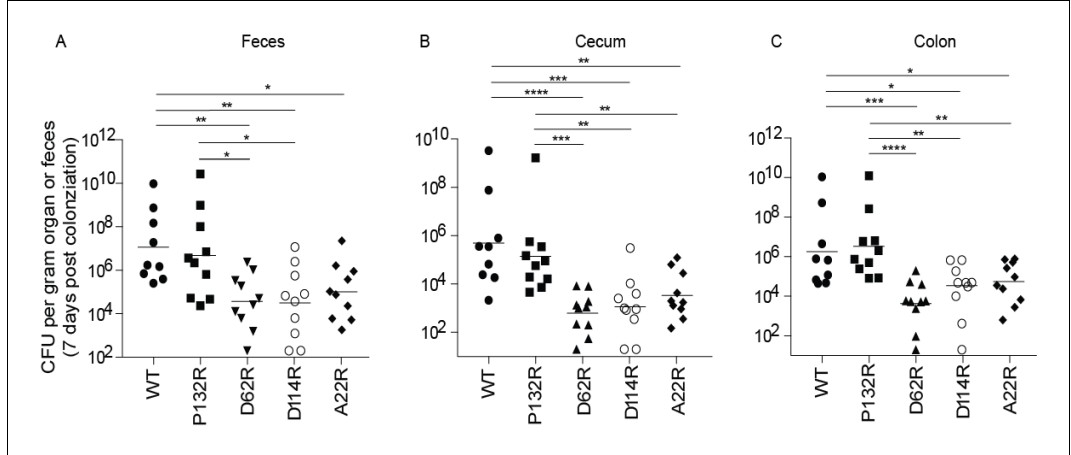

**Figure 6.** Mutations in FimA alter the ability of UTI89 to colonize the intestine. C3H/HeN mice were orally gavaged with streptomycin, to disrupt colonization resistance, and one day later orally gavaged with $1 \times 10^8$ CFU of WT UTI89 or one of 4 isogenic UTI89 strains with point mutations in *fimA*. The burden of each FimA mutant was determined via CFU counting in were determined the feces, cecum, or colon of mice at 7 days post colonization. Bars represent geometric mean. *p<0.05, **p<0.01, ***p<0.001, ****p<0.0001 by Mann Whitney U test. N = 10, 2 biological replicates.

DOI: https://doi.org/10.7554/eLife.31662.018

pathogen interactions. We identified three mutations in FimA$_{UTI89}$ (A22R, D114R, and D62R) that expressed adhesive pili in vitro but reduced the ability of UTI89 to colonize the bladder, acutely and chronically. One of these mutants, FimA$_{UTI89}$ A22R, was severely attenuated during acute infection, forming almost no IBCs and thus was unable to chronically infect the mouse bladder.

## The impact of FimA mutants on the 'molecular-spring' function of the pilus rod

We hypothesize that the identified FimA point mutants handicap UPEC pathogenicity in vivo by altering the properties of the pilus rod. The pilus rod is hypothesized to act as a 'molecular spring' transitioning between a flexible, linear fiber and a coiled helix. This spring-like property is thought to prevent the pilus from breaking or detaching from the host surface by temporarily expanding to a linear form after encountering shear forces, which can occur in the bladder during urine voiding or in the gut during fecal or mucus shedding. Such a transition between the helical and unwound form of the FimA homopolymer has been predicted to significantly dampen the force experienced by the adhesion-receptor complex at the tip (*Zakrisson et al., 2012*). Since FimA-FimA interactions create the bulk of the pilus rod, mutations that reduce the stability of these protein-protein interactions can alter the ability of the pilus rod to withstand shear forces, which we discovered has detrimental effects on pilus function and pathogenesis. This is consistent with ~50% of the residues in the mature FimA protein (79/161) being invariant (*Figure 3*), which includes almost every residue involved in a subunit-subunit interface in the rod. Pili formed with the WT FimA$_{BW25113}$ protein require more than twice the force to unwind than the pili formed by the FimA $_{BW25113}$ variant with an A22R mutation (~30 pN vs. 11pN), likely due to steric clashes caused by this substitution affecting the tightly packed interface between subunits. Thus, we hypothesize that the pilus formed by the FimA A22R mutant variant may not have the strength to absorb and withstand the shear forces experienced during urination and thus allow the bacteria to be swept out of the bladder, preventing infection. In the gut, the pilus rod may play a similar role during mucus shedding. A recent study suggests that type 1 pili may promote UPEC colonization of the upper crypts (*Spaulding et al., 2017*). These bacteria would likely experience constant, low levels of shear force during mucus turnover and thus need the pilus to withstand some force to enable the bacteria to maintain their intestinal niche. Accordingly, three of the *fimA$_{UTI89}$* mutants (A22R, D114R, and D62R) that were attenuated in bladder colonization also displayed significantly reduced intestinal colonization. However, in general, the phenotypes of the *fimA$_{UTI89}$* mutants in gut colonization were not as severe as in the bladder.

Two mutations in FimA$_{UTI89}$ (D62R and D114R) resulted in a mild clumping phenotype when bacteria were grown in vitro. This clumping phenotype was also seen in the equivalent mutants in FimA$_{BW25113}$ (D62R, D114R). These residues are located on the exterior surface of the rod structure and do not have as strong an effect on the force needed for unwinding as FimA$_{BW2511}$ A22R. However, the clumping phenotype suggests that these mutations may alter side-to-side interactions between different pilus rods promoting pilus-pilus interactions within and between bacteria, thus inhibiting the pilus mediated interactions needed in vivo for attachment, invasion and/or IBC formation. Interestingly, we do not observe a defect in bladder colonization in mice infected with FimA$_{UTI89}$ D62R until 10dpi, suggesting that normal rod function is needed during both chronic and acute infection.

Together, our data suggest that the type 1 pilus rod may mediate colonization phenotypes through damping of shear forces or through other mechanism(s) outside of scaffold support for FimH-binding. A possibility that we cannot exclude at the moment is that the lower uncoiling force of the A22R mutant exposes parts of the rod that would be largely buried in the wt filament to host proteases. This would still suggest that the mechanical properties of these rods have been 'tuned' to optimize for particular environments, as a wt rod that could never uncoil would be susceptible to breakage by the shear forces, while a rod that uncoils too easily would be unnecessarily exposed to digestion by proteases.

Our analyses indicate that FimA is undergoing selection pressures due to as-yet undefined host-pathogen interactions, which may explain some colonization defects seen here. FimA displays patterns of episodic, divergent selection on surface exposed amino acids in a pattern that is similar, though much less robust, to what is seen in the flagellin protein in *Salmonella* (*Li et al., 1994*). In *Salmonella*, the flagellin subunits are split into several domains where the domains responsible for subunit-subunit assembly are highly conserved while the more external domains of the protein show

high rates of variability (*Andersen-Nissen et al., 2005*; *Galkin et al., 2008*). The *Salmonella* flagellin protein is immunogenic, but the regions that induce inflammatory response are found in the conserved domains of the protein (*Wildschutte et al., 2004*). While it is known that flagellin are bound by Toll-like receptor 5 (TLR5) resulting in the induction of the innate immune response (*Smith and Ozinsky, 2002*) and that this recognition is targeted towards conserved features of flagellin (*Andersen-Nissen et al., 2005*), it is currently unknown which, if any, host immune receptor are capable of recognizing the FimA rod or subunits or which structural features are targeted by the host immune system. Further research is needed to fully elucidate the host pressures and responses that are influencing the evolution of the FimA rod structure.

## The evolution of the type 1 pilus rod

Evolutionary and structural analysis of FimA, in combination with our in vitro and in vivo phenotyping, yielded several important insights into the selection pressures faced by UPEC as well as the evolutionary trajectories that pathogens follow to enhance their colonization of different host niches. Notably, we saw that different phenotypes caused by mutations in the FimA protein were associated with different classes of selection pressure. For example, the bacterial clumping phenotype is associated with mutation of three codons under purifying selection, but an equal number of the mutations that resulted in clumping are in codons with little to no evidence of selective pressure. In contrast, all mutations that failed to complement a *fimA* gene deletion were made in codons that are under very strong purifying selection or are too conserved for analysis. Together, the difference in selection pressure suggests that the mis-assembly of the pilus is much more harmful and/or toxic to *E. coli* than bacterial clumping. Further, the integration of evolutionary analysis with in vivo and in vitro functional analysis allowed us to decouple the selection pressures acting to preserve amino acid sites necessary for pilus assembly (such as P145in FimA$_{UTI89}$) from the selection pressure maintaining the codons that were necessary for pilus function (such as A22 in FimA$_{UTI89}$). Given the intricacy of pilus assembly, one could expect that most of the codons under purifying selection would be related to pilus construction. Instead, we found that many codons under purifying selection are involved in keeping the force needed to unwind the type 1 rod within a narrow range. This leads to reasoning that a 'weak' or 'loose' FimA rod may be just as detrimental for *E. coli* as having no rod at all, at least in the eyes of evolution.

In summary, by combining structural studies, force spectroscopy, genetic analysis, and relevant mouse models of UTI and gut colonization, we conclude that the mechanical properties of the type 1 pilus rod are essential for its functional role in mediating *E. coli* pathogenesis and persistence and appear to have been carefully 'tuned' by evolution. Further studies of the hundreds of CUP pili encoded in Gram-negative bacteria are needed to further understand the unique and general aspects of the evolution of CUP pilus fibers. In addition, other bacterial pili, such as T4P, which have arisen independently of CUP pili but can play similar roles in pathogenesis, can also elongate under force (*Biais et al., 2010*) and thus it remains an interesting question as to how the physical properties of other pili have been selected for particular environments and how these properties impact bacterial pathogenesis.

## Materials and methods

### Ethics statement

The Washington University Animal Studies Committee approved all procedures used for the mouse experiments described in the present study. Overall care of the animals was consistent with *The Guide for the Care and Use of Laboratory Animals* from the National Research Council and the USDA *Animal Care Resource Guide*.

### Bacteria, cloning, mutagenesis

The BW25113 *fimA* gene sequence was cloned between the *EcoRI* and *BamHI* restriction sites in pTRC99A using standard PCR cloning techniques to create plasmid pTRC-fimA. Mutations were made within this plasmid using appropriate complementary primers to engineer codon changes in the template, pTRC-fimA, using *Pfx* polymerase and manufacturers instructions for PCR, followed by *DpnI* treatment of the resulting products to remove the methylated template before transformation

into C600. Mutations were verified by sequencing. Mutant plasmids were transformed into UTI89-LON, ΔfimA for expression and functional studies as indicated.

In order to construct point mutations in the *fimA* allele in the UTI89 chromosome, the UTI89 *fimA* gene was deleted using a previously published technique that allows for flawless integration (*Khetrapal et al., 2016*). Briefly, *fimA* was deleted by homologous recombination using pSLC- 217 as a template and primers containing 50 bp of homology to flanking regions of *fimA*. A deletion was then constructed using the previously described Red Recombinase method that would allow for reinsertion of constructs into the *fimA* site. Concurrently, a copy of UTI89 *fimA* was cloned into pTRC99a. Point mutations were then introduced into this construct using site directed mutagenesis. PCR fragments from confirmed mutants, and the WT allele, were reintegrated into the UTI89-LON, ΔfimA mutants constructed above at the original deletion site. Successful reintegration events were sequenced to confirm flawless integration and mutation presence.

## Mouse studies

Animals were maintained in a single room in our vivarium. Prior to and after infection all animals received PicoLab Rodent Diet 20 (Purina) *ad libitum*. All animals were maintained under a strict light cycle (lights on at 0600 hr, off at 1800 hr). Mice were acquired from indicated vendors and randomly placed into cages (n = 5 mice/cage) by employees of Washington University's Division of Comparative Medicine (DCM); no additional methods for randomization were used to determine how animals were allocated to experimental groups. Investigators were not blinded to group allocation during experiments.

For bladder infections, 6 week old female C3H/HeN mice were obtained from Envigo and were maintained in our vivarium for one week prior to infection. Bladder infections were performed via transurethral inoculation (*Hung et al., 2009*). UPEC strains were prepared for inoculation as described previously (*Hung et al., 2009*). Briefly, a single UTI89 colony was inoculated in 20 mL of Luria Broth (LB) and incubated at 37°C under static conditions for 24 hr. Bacteria were then diluted (1:1000) into fresh LB and incubated at 37°C under static conditions for 18–24 hr. Bacteria were subsequently washed three times with PBS and then concentrated to ~1×10$^8$ CFU per 100 μL for intestinal infections and ~1×10$^8$ CFU per 50 μL for bladder infections. Bacteria were subsequently washed three times with PBS and then concentrated to ~1×10$^8$ CFU per 50 μL for bladder infections.

For intestinal colonization experiments, 6 week old female C3H/HeN mice were obtained from Envigo and were maintained in our vivarium for no more than 2 days prior to intestinal colonization. Mice received a single dose of streptomycin (1000 mg/kg in 100 μL water by oral gavage (PO)) followed 24 hr later by an oral gavage of ~10$^8$ CFU UPEC in 100 μL phosphate-buffered saline (PBS) (*Spaulding et al., 2017*). Bacteria were subsequently washed three times with PBS and then concentrated to ~1×10$^8$ CFU per 100 μL for intestinal infections.

In all cases, fecal and urine samples were collected directly from each animal at the indicated time points. Fecal samples were immediately weighed and homogenized in 1 mL PBS. Urine samples were immediately diluted 1:10 prior to plating. Mice were sacrificed via cervical dislocation under isofluorane anesthesia and their organs were removed and processed under aseptic conditions. Intestinal segments (cecum and colon) were weighed prior to homogenization and plating on LB supplemented with the appropriate antibiotic.

Exclusion criteria for mice were pre-established; (i) both introduced strains in competitive infections became undetectable during the course of a 14 day experiment, and (ii) mice died or lost >20% of their body weight. No mice in this study met these criteria. Each experiment was conducted with both technical (i.e., a single inoculum of bacteria) and biological (i.e., separate bacterial cultures of the same strain) replicates.

## Enumeration of intracellular bacteria

6 week old female C3H/HeN mice were given a transurethral inoculation with WT UTI89 or a *fimA* mutant strain. To accurately count the number of IBCs, mice were sacrificed 6 or 12 hr after infection. Bladders were removed aseptically, bi-sected, splayed on silicone plates and fixed in 4% (v/v) paraformaldehyde. IBCs, readily discernable as punctate violet spots, were quantified by LacZ staining of bladder wholemounts (*Justice et al., 2006*; *Cusumano et al., 2011*). Bacterial invasion assays were performed at 1, 3, and 6 hr post infection as previously described (*Mulvey et al., 1998*).

## Hemagglutination assays (HA)

Bacteria were grown under type 1 pilus-inducing conditions (*Greene et al., 2015*), with appropriate antibiotics and. 01-.02mM IPTG induction, if indicated. Pilus expression was assessed by hemagglutination assays (HA) as previously described (*Greene et al., 2015*) using bacterial cultures normalized to an optical density at 600 nm ($OD_{600}$) of 1 and guinea pig erythrocytes normalized to an $OD_{640}$ of 2. The experiment was conducted in parallel in PBS with 2% w/v methyl-α-D-mannopyranoside.

## Electron microscopy

Electron micrographs (EM) were taken of UTI89 or UTI89 isogenic mutants after growth under type 1 pilus-inducing conditions. A total of 300 bacterial cells were counted for each condition, and piliation on those cells was classified as bald (no pili), low (1 to 20 pili/cell), moderate (20 to 200 pili/cell), or abundant (>200 pili/cell).

## Force extension experiments

For expression of type 1 pili the strains were grown in Luria Broth (LB) supplemented with carbenicillin (100 µg/mL) and IPTG (50 µM), at 37°C overnight. The optical tweezers (OT) setup is built around an inverted microscope (Olympus IX71, Olympus, Japan) equipped with a high numerical aperture oil immersion objective (model: UplanFl 100X N.A. = 1.35; Olympus, Japan) and a 1292 × 964 pixel camera with a cell size of 3.75 × 3.75 µm (model: StingRay F-125, Allied Vision) (*Mortezaei et al., 2013*)(*Figure 4—figure supplement 1*). The OT stands in a temperature controlled room with computers and controllers isolated from the room to reduce noise and vibrations. We use a continuous wave Nd:YVO$_4$ laser (Millennia IR, Spectra Physics, Santa Clara, CA) operating at 1064 nm for trapping a single bacterium or microspheres. A probe laser (low power HeNe-laser operating at 632.8 nm) is merged with the trapping laser using a polarizing beam splitter cube (PBSC). The light from the probe laser is refracted by the trapped object and collected by the condenser and thereafter imaged onto a 2D position sensitive detector (PSD, L20 SU9, Sitek Electro Optics, Sweden). The PSD convert the incoming light to a photocurrent and thereafter to a voltage that is sent to a programmable low pass filter (SR640, Stanford research systems), later collected by a computer and processed with an in-house LabVIEW program. We minimized the amount of noise in the setup and optimized the measured time series using the Allan variance method described in (*Andersson et al., 2011*).

To prepare a sample we suspended bacteria in 1xPBS to a concentration (1:1000 of $OD_{600}$ = 1) suitable for single cell analysis using optical tweezers (OT). Surfactant-free 2.5 µm amidine polystyrene microspheres (product no. 3–2600, Invitrogen, Carlsbad, CA) were similarly suspended in Milli-Q water, these microspheres were trapped and used as force probes. To mount bacteria and reduce the influence of surface interactions we prepared a 1:500 suspension of 9.5 mm carboxylate-modified latex microspheres (product no.2–10000, Interfacial Dynamics, Portland, OR) in Milli-Q. We dropped ten microliters of the microsphere-water suspension onto 24 × 60 mm coverslips (no.1, Knittel Glass, Braunschweig, Germany) and placed these in an oven for 60 min at 60°C to immobilize the microspheres to the surface. To firmly adhere bacteria to the microspheres, we added a solution of 20 mL of 0.01% poly-L-lysine (catalog no. P4832, Sigma-Aldrich, St. Louis, MO) to the coverslips, which, after 45 min incubation at 60°C, were stored until use. A free-floating bacterium was trapped by the optical tweezers run at low power to avoid cell damage. The bacterium was thereafter mounted on a large 9.5 µm microsphere coated with poly-L-lysine. We trapped a small free-floating 2.5 µm microsphere by the optical tweezers with normal power (a few hundreds of mW) and brought it close to (within tens of µm) but not in direct contact with, the bacterium. To calibrate the trap stiffness we used the Power spectrum method by sampling the microspheres position at 131,072 Hz and average 32 consecutive data sets acquired for 0.25 s each (*Tolić-Nørrelykke et al., 2006*). Typically, the trap constant was found to be ~140 pN/µm for an output laser power of 800 mW. After calibration, the small microsphere was gently brought close to the bacterium in order to attach a pilus with the microsphere (*Figure 4—figure supplement 1*). To extend a single pilus (*Figure 4*) the piezo stage was moved at a constant speed of 10 nm/s and the sampling frequency was set to 10,000 Hz that was downsampled by 800. Occasionally, we measured the responses of multiple pili attached to the bead resulting in a force-extension response as the sum of all attached pili. This was, however, not a problem in general since the shorter pili detached from the bead in a sequential order, leaving

only the single, longest pili attached to the bead for measurement (*Figure 4—figure supplement 2*). Finally, we controlled the piezo-stage and sampled the data using an in-house LabView program (*Andersson, 2018*; copy archived at https://github.com/elifesciences-publications/ot-control).

To make a flow chamber, we added a ring of vacuum grease (Dow Corning, Midland, MI) around the area containing the poly-L-lysine-coated microspheres on one of the coverslips. Carefully, we dropped a 3 mL suspension of bacteria and a 3 mL suspension of probe microspheres (surfactant-free 2.5 mm white amidine polystyrene latex microsphere, product no. 3–2600, Invitrogen, Carlsbad, CA) onto the area and sealed the flow chamber by placing a 20 × 20 mm coverslip (no.1, Knittel Glass) on top. Thereafter, we mounted the sample in a sample holder that is fixed to a piezo-stage (Physik Instrument, P-561.3CD stage) in the OT instrumentation. To get a reliable OT calibration parameter values we measured the temperature using a thermocouple in the sample chamber, 23.0 ± 0.1°C and the suspension viscosity was assumed to only vary with temperature, thus, the viscosity was set to 0.932 mPas ± 0.002 mPas.

## Pilus preparation for structural determination

To prepare the pilus extracts, bacteria of the *E. coli* strain BW25113 (*Datsenko and Wanner, 2000*) were inoculated by dense streaking on eight M9 minimal agar plates containing 0.5% glycerol (vol/vol). After a 72 hr incubation at 30°C, bacteria were harvested in 30 mL of LB medium, vortexed vigorously for 5 min and passed eight times through a 26-Gauge needle, to detach pili from the cells. Bacteria were removed at 4°C by three successive 10 min centrifugation steps at 16,000 x g. To collect the pili, cleared supernatants were centrifuged for 1 hr at 100,000 *g* in a cold Beckman Ti60 ultracentrifuge rotor. Pellet containing the crude pilus fraction was taken up in 200 μL of 50 mM HEPES, 50 mM NaCl pH 7.4, and maintained at 4°C for further analysis.

## Cryo-EM data collection and image processing

3 μL of sample was applied to glow discharged lacey carbon grids (TED PELLA, Inc., 300 mesh). Then the grids were plunge-frozen using a Vitrobot Mark IV (FEI, Inc.), and subsequently imaged in a Titan Krios at 300keV with a Falcon II direct electron detector (pixel size 1.05 Å/pixel). A total of 6803 images, each of which was from a total exposure of 2 s dose-fractionated into seven chunks, were collected at a range of underfocus between 0.5 ~ 3 μm. Images were motion corrected using MotionCorr (*Li et al., 2013*), and the program CTFFIND3 (*Mindell and Grigorieff, 2003*) was used for determining the defocus and astigmatism. Images with poor CTF estimation as well as defocus >3 μm were discarded. The SPIDER software package (*Frank et al., 1996*) was used for most other operations with the first two-chunk sums (containing a dose of ~20 electrons/ Å$^2$) of the motion-corrected image stacks. The CTF was corrected by multiplying the images from the first two-chunk sums with the theoretical CTF, which is a Wiener filter in the limit of a very poor signal-to-noise ratio (SNR). This both corrects the phases which need to be flipped and improves the SNR. The e2helixboxer routine within EMAN2 (*Tang et al., 2007*) was used for boxing the filaments from the images. A total of 72,627 overlapping segments (384 px long), with a shift of 11 px between adjacent segments (~97% overlap), were used for the IHRSR (*Egelman, 2000*) reconstruction. With a featureless cylinder as a starting reference, 72,627 segments were used in IHRSR cycles until the helical parameters (axial rise and rotation per subunit) converged. Analysis of the population suggested that the axial rise was fairly fixed, but that the twist was variable. Using a reference-based sorting with models having a fixed rise but a variable twist, approximately 55% of the segments were excluded, having a twist outside of the range 114.4° to 115.6°. A sub-set of 32,726 segments were used for a few more cycles of IHRSR. The resolution of the final reconstruction was determined by the FSC between two independent half maps, generated from two non-overlapping data sets, which was 4.2 Å at FSC = 0.143.

## Model building and refinement

We used a previous FimA NMR model (PDB id: 2JTY, a single chain) as an initial template to dock into the cryo-EM map by rigid body fitting, and then manually edited the model in Chimera (*Pettersen et al., 2004*) and Coot (*Emsley et al., 2010*). We then used the combined model (1–19 and 21–159) as the starting template to re-build a single chain of the FimA protein using the RosettaCM protocol (*Wang et al., 2015*). Next, the full length model of FimA missing the first two residues and the last

residue was iteratively refined by Phenix real-space refine (*Adams et al., 2010*) and manually adjusted in Coot. The refined single chain of the FimA model was then re-built by RosettaCM (*Wang et al., 2015*) with helical symmetry and refined by Phenix to improve the stereochemistry as well as the model map coefficient correlation. The FimA model was validated with MolProbity (*Chen et al., 2010*) and the coordinates deposited to the Protein Data Bank with the accession code 6C53 (atomic structure). The corresponding cryo-EM map was deposited in the EMDB with accession code EMD-7342. The refinement statistics are given in *Supplementary file 1*.

## Bioinformatic analyses

The protein sequences of closely-related homologs of *E. coli* FimA, FimC, and FimH were obtained by individual searches the Ensembl Bacteria Genome database (*Kersey et al., 2016*) using the phmmer web-server (*Finn et al., 2015*) with a BLOSUM62 (FimA) or a BLOSUM90 (FimC and FimH) scoring matrix using the full-length *E. coli* UTI89 protein sequences as queries. The sequence matches were then filtered to remove low scoring hits and non-functional sequences (i.e., those predicted to lack critical sequence features such as complete signal sequences and/or C-terminal tyrosine residues in FimA and FimH). The signal sequences were trimmed from the remaining homologs using Geneious v 6.1.7 (*Kearse et al., 2012*) and the protein sequences were aligned with the MAFFT program using two iterations of the FFT-NS-i algorithm based on the PAM200 scoring matrix (*Katoh and Standley, 2013*). Conservation at each amino acid position was calculated using custom Python scripts and a sequence logo was created using the ggseqlogo package in R (*R Core Team, 2017*) using RStudio (*RStudio Team , 2015*).

To estimate selection pressures on each codon in *fimA*, we obtained all available gene sequences encoding the protein sequences described above from the Ensembl Bacteria Genomes database (n = 1,825, three were removed by submitter's request) using custom bash scripts (*Supplementary file 3*). A total of 191 unique sequences were identified using Geneious v 6.1.7 and trimmed to remove the signal sequence. Evolutionary model selection was performed using maximum likelihood ratio testing on the Datamonkey webserver (*Pond and Frost, 2005*; *Delport et al., 2010*), which identified the TIM2 model (model 010232) as the most likely model of nucleotide substitution for the *fimA* homologs. These sequences were then were scanned for evidence of recombination using single breakpoint analysis (*Kosakovsky Pond et al., 2006*) and phylogenetic trees with a correction for a breakpoint found in codon 107 (position 320) were generated. These phylogenetic trees and evolutionary model were then used to measure the ratio of the rates non-synonymous (*dN*) to synonymous (*dS*) mutation in each codon (i.e., a *dN/dS* ratio) using a fixed-effects likelihood test to identify statistical significance (*Kosakovsky Pond and Frost, 2005*).

Using a collection of 67 curated, reference genomes from a previous study (*Schreiber et al., 2017*), we examined the carriage and phylogenetic context of *fimA* carriage using a BLAST-based search (*Camacho et al., 2009*) with the UTI89 *fimA* gene as a query. Full-length sequences were extracted from the genomes using Geneious v 6.1.7, trimmed to remove signal sequences, and aligned using the MUSCLE program (*Edgar, 2004*). A phylogenetic tree was estimated using the RAxML program (*Stamatakis, 2006*) with the GTRCAT model and supported with 1000 bootstraps (*Stamatakis et al., 2008*). Evidence for episodic, diversifying selection was then identified using a random effects likelihood ratio test for each branch of the *fimA* phylogenetic tree (*Kosakovsky Pond et al., 2011*) using unique sequences from the genomes (32 duplicates removed, n = 25). Branches showing statistically significant evidence for episodic, diversifying selection, as measured by a chi-squared test were then indicated on the corresponding branches of the phylogenetic tree constructed using RAxML.

## Acknowledgements

We would like to thank Wandy Beatty and Bryan Anthony of the WUSM Molecular Microbiology Imaging Facility. This work was supported by grants from the National Institutes of Health [GM122510 (EHE), AI048689 and DK064540 (SJH), 1F31DK107057 (CNS), and DK101171-02 (MSC)], the Swedish Research Council 621-2013-5379 (MA) and the Agence Nationale de la Reserche ANR-14-CE09-0004 (OF). ALR was funded by the Pasteur Paris University PhD program. The cryo-EM work was conducted at the Molecular Electron Microscopy Core facility at the University of Virginia, which is supported by the School of Medicine and built with NIH grant G20-RR31199. The Titan Krios and Falcon II direct electron detector within the Core were purchased with NIH SIG S10-RR025067 and S10-OD018149, respectively.

## Additional information

### Competing interests

Edward H Egelman: Reviewing editor, *eLife*. The other authors declare that no competing interests exist.

### Funding

| Funder | Grant reference number | Author |
|---|---|---|
| National Institutes of Health | GM122510 | Edward H Egelman |
| National Institutes of Health | AI048689 | Scott Hultgren |
| National Institutes of Health | DK064540 | Scott Hultgren |
| National Institutes of Health | 1F31DK107057 | Caitlin N Spaulding |
| National Institutes of Health | DK101171-02 | Matt S Conover |
| Svenska Forskningsrådet Formas | 621-2013-5379 | Magnus Andersson |
| Agence Nationale de la Recherche | ANR-14-CE09-0004 | Olivera Francetic |
| Paris Pasteur University | Graduate Research Fellowship | Areli Luna-Rico |
| Washington University School of Medicine in St. Louis | Lucille P. Markey Pathway for Pathobiology | Henry Louis Schreiber IV |
| Washington University School of Medicine in St. Louis | Monsanto Excellence Fund Graduate Fellowship | Henry Louis Schreiber IV |

The funders had no role in study design, data collection and interpretation, or the decision to submit the work for publication.

### Author contributions

Caitlin N Spaulding, Formal analysis, Investigation, Visualization, Methodology, Writing—original draft, Writing—review and editing; Henry Louis Schreiber IV, Data curation, Formal analysis, Investigation, Visualization, Methodology, Writing—original draft, Writing—review and editing; Weili Zheng, Data curation, Formal analysis, Investigation, Visualization, Writing—original draft, Writing—review and editing; Karen W Dodson, Conceptualization, Formal analysis, Supervision, Investigation, Visualization, Methodology, Writing—original draft, Project administration, Writing—review and editing; Jennie E Hazen, Formal analysis, Investigation; Matt S Conover, Fengbin Wang, Investigation, Visualization, Writing—review and editing; Pontus Svenmarker, Data curation, Investigation, Visualization, Writing—review and editing; Areli Luna-Rico, Resources, Investigation, Writing—review and editing; Olivera Francetic, Resources, Supervision, Investigation, Methodology, Writing—review and editing; Magnus Andersson, Resources, Supervision, Funding acquisition, Investigation, Visualization, Methodology, Writing—review and editing; Scott Hultgren, Conceptualization, Resources, Formal analysis, Supervision, Funding acquisition, Methodology, Writing—original draft, Project administration, Writing—review and editing; Edward H Egelman, Conceptualization, Resources, Formal analysis, Supervision, Funding acquisition, Visualization, Methodology, Writing—original draft, Project administration, Writing—review and editing

### Author ORCIDs

Caitlin N Spaulding (iD) http://orcid.org/0000-0002-7582-3816
Henry Louis Schreiber IV (iD) http://orcid.org/0000-0002-4501-9886
Pontus Svenmarker (iD) http://orcid.org/0000-0002-1308-4923
Areli Luna-Rico (iD) http://orcid.org/0000-0001-7538-5441
Olivera Francetic (iD) http://orcid.org/0000-0002-4145-5314
Magnus Andersson (iD) http://orcid.org/0000-0002-9835-3263

Scott Hultgren [ID] http://orcid.org/0000-0001-8785-564X
Edward H Egelman [ID] http://orcid.org/0000-0003-4844-5212

### Ethics

Animal experimentation: The Washington University Animal Studies Committee approved all procedures used for the mouse experiments described in the present study (Protocol Application Number 20150226). Overall care of the animals was consistent with The Guide for the Care and Use of Laboratory Animals from the National Research Council and the USDA Animal Care Resource Guide. Every effort was made to minimize suffering.

### Decision letter and Author response

Decision letter https://doi.org/10.7554/eLife.31662.030
Author response https://doi.org/10.7554/eLife.31662.031

## Additional files

### Supplementary files

• Supplementary file 1. Validation statistics for FimA model
DOI: https://doi.org/10.7554/eLife.31662.019

• Supplementary file 2. Helical parameters comparison within FimA models
DOI: https://doi.org/10.7554/eLife.31662.020

• Supplementary file 3. FimA sequences used in analysis of conservation and selection
DOI: https://doi.org/10.7554/eLife.31662.021

• Supplementary file 4. List of FimC and FimH sequences
DOI: https://doi.org/10.7554/eLife.31662.022

• Supplementary file 5. Codon-by-codon selection analysis in *fimA*
DOI: https://doi.org/10.7554/eLife.31662.023

• Supplementary file 6. Reference and UPEC strains used in analysis of *fimA* carriage
DOI: https://doi.org/10.7554/eLife.31662.024

• Supplementary file 7. In vivo and in vitro phenotypes of *fim A* mutants
DOI: https://doi.org/10.7554/eLife.31662.025

• Transparent reporting form
DOI: https://doi.org/10.7554/eLife.31662.026

### Major datasets

The following previously published dataset was used:

| Author(s) | Year | Dataset title | Dataset URL | Database, license, and accessibility information |
|---|---|---|---|---|
| Schreiber IV HL, Conover MS, Chou WC, M Hibbing E, Manson AL, Dodson KW, Hannan TJ, Roberts PL, Stapleton AE, Hooton TM, Livny J, Earl AM, Hultgren SJ | 2017 | Bacteria Genome sequencing and assembly | https://www.ncbi.nlm.nih.gov/bioproject/?term=PRJNA269984 | Publicly available at the NCBI BioProject Database (accession no:PRJNA269984). |

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
