## [Decision Letter]

Thank you for submitting your article "Functional role of the type 1 pilus rod structure in mediating host-pathogen interactions" for consideration by *eLife*. Your article has been reviewed by three peer reviewers, including Michael Gilmore as Reviewing Editor, and the evaluation has been overseen by Gisela Storz as the Senior Editor.

We have drafted this decision to help you prepare a revised submission.

Summary:

This is a well-written structural biology study that employs state-of-the art cryo EM to solve the structure of the assembled type 1 pilus rod of *E. coli*. This pilus is a major colonization factor for *E. coli* and other gram-negative microbes, in the gut and in the bladder, with a well-established role in urinary tract infection. Prior to this study, substantial knowledge of the type 1 pilus structure derived from multiple approaches to solve the structure of the FimA monomer, and NMR data on polymerized FimA. The data obtained here extends and modifies the understanding of its structure in several specific aspects. Moreover, force measurements of mutant variants of the pilus are consistent with a model that proposes that specific extension forces are functionally important, and the tension of its spring-like quality is critical for causing disease. Additionally, useful insights are made into the evolution of variants of this structure, and selective forces that may influence that. All of the elements of the study appear to be well designed and executed, and reflect both a deep understanding of the subject of study, and advanced technologies for its exploration.

Essential revisions:

There are two major points to be addressed:

1) Determination of unwinding force and correlation between unwinding force and in vivo observations are not convincing yet and need to be substantiated.

2) When correlating the in vitro and in vivo results, there are several uncontrolled variables that need to be addressed including proteases, bacterial clumping, number of pili versus structure of pili for fimA mutants.

Further detail can be found in the separate reviews below. If you have any questions about the revisions, please let us know.

*Reviewer #1:*

The only concern of this reviewer is that some leaps are made to correlate observations in vitro, with those in vivo. The suggestion that the spring-like qualities of the pilus rod are critical for pathogenesis is entirely possible, but is far from proven. Moreover, few specifics are provided that relate the actual shear forces of urine at various distances from the mucosal boundary to the forces measured for variations in the protein. A number of uncontrolled variables could explain why mutant variants of the protein are less effective in vivo, including increased susceptibility to tissue resident proteases, altered interactions with various elements of the mucosa, altered physical states imposed (such as the clumping that was observed), etc. While an interesting prospect, the mechanical properties proposed could be explored further for proof, or the discussion of this prospect modified in the text (although the latter arguably could affect the perceived impact of the study in altering existing paradigms). Aside from that concern, this reviewer found the paper to be of considerable interest and well done.

*Reviewer #2:*

The authors determined the Type I pilus structure by cryo electron microscopy. The images in Figure 1 are stunning. The determine force required to unwind a single FimA rod, and then introduce mutations that would be predicted to destabilize the wound pilus structure. These mutant pili require less force to unwind. Interestingly, UPEC expressing these mutant pili were significantly attenuated in bladder infection and intestinal colonization in mice, despite being able to agglutinate red blood cells in vitro. The authors have not identified the specific host immune response which contribute to pathogenic deficiencies, but clearly the sequence of the FimA major subunit are critical for pilus function in the bladder and gut and thus has a major impact on the outcome of UTI. Below are some comments and suggestions that could be addressed in a revised manuscript.

1) The A22R mutant has a significant defect at 6 hours in the mouse. Does this mutant affect the ability of bacteria to invade the superficial facet cells?

2) A few of the mutants were tested in vivo, but not in vitro. For example, E123R was tested in the mouse and displayed a defect in acute and chronic UTI (and a slight defect in gut colonization), but the force measurements for unwinding this pili were not determined.

*Reviewer #3:*

The manuscript by Caitlin Spaulding et al. reports on the relation between atomic structure, evolutionary conservation, biophysical properties, and host pathogen interactions of bacterial type I pili. Using electron microscopy, they solve the structure of the type I pilus at atomic resolution. Next, by genomic comparison of the T1P genes between different species, they find that the gene encoding of the major subunit of the pilus, fimA, shows a higher variability compared to other parts of the T1P machinery. In particular, residues at the exterior are prone to adaptive selection. Based on their positions in the atomic model, the authors investigate the effects of point mutations on pilus expression and agglutination. They state that the unwinding force of the pilus strongly depends on these mutations while functionality is unaffected. Finally, they investigate the effect of point mutations on the competitive fitness within the bladder of a mouse model, on their ability cause chronic infections, and on their ability to colonize the intestine. The authors conclude that reducing the unwinding force of T1P affects host-pathogen interactions.

The idea to test the relevance of elastic properties of pili for host-pathogen interactions and in particular of bladder infections is wonderful. The general approach of starting with an atomic model of the pilus to identify promising candidates for amino acid substitutions is highly convincing. I have three major problems with this study and I will describe them in detail below. I would like to state that I don't have enough expertise in structural biology to assess the quality of the structural data.

1) Measurement of the unwinding force.a) The authors state in the Materials and methods part they expect "attachment a few pili with the microsphere". Determination of the unwinding force, however, is only possible it the authors can make sure that only a single pilus is attached at the time.

b) I haven't found any information about how the mutations analysed in Figure 4 affect the number of pili per cell. If more than a pilus is bound during a force measurement, then the level of piliation can affect the measured uncoiling force.

c) Only technical replicates from a single sample are inacceptable. Data should be acquired on biological replicates on at least three different days.

2) Expression levels and functionality of pili with point mutations.

The authors state that "FimA mutant strain produced minimal changes to the levels of type 1 piliation […] and to the function of the type 1 pilus". The experimental data supporting this statement is shown in Figure 5—figure supplement 1. I cannot understand this statement based on the data. The percentage of "bald" bacteria is about 3x higher for the D116R strain compared to wt. The mannose-inhibitable HA-titre is about 10x higher in the E123R and D116R strains compared to wt. We can expect that the pilus density affects host-pathogen interactions. One way of showing that the 3x increase of pilus density does not affect the conclusion, would be to generate strains with wt fimA sequence but decreased pilus levels.

3) Comparison between different strains.a) The authors explain that they used different bacterial strains with different fimA sequences. This is probably fine, but the way the data is presented now, I get confused with the different notations of the point mutations. For example, is the A22R mutation in Figure 4 comparable to A24R in Figure 5? Same for D114R and D116R etc.? Can the authors come up with a smart way of indicating these connections within the figures?

b) The authors look at the effects of amino acid substitutions at different levels of complexity (which is great) but I currently get lost in the details. Please explain the criteria for analysing specific mutations better within the individual parts of the Results section. Also, please explain the connections between the conclusions drawn from mutant analysis of conservation, unwinding, and host-pathogen interactions related to the structure in an explicit way.

---

## [Author Response]

Essential revisions:There are two major points to be addressed:1) Determination of unwinding force and correlation between unwinding force and in vivo observations are not convincing yet and need to be substantiated.

We thank the reviewers for bringing up this point. Determination of the unwinding force of the WT type 1 pili in this study correlates well with previously published studies (Andersson et al., 2007; Jass et al., 2004; Whitfield et al., 2014). There have been numerous parallel-plate flow chamber studies mimicking the fluid conditions exposing bacteria attaching to host cells. These studies show indeed the need of pili for cell attachment (works by Wendy Thomas group). In such in vitro studies, it has been observed that pili do in fact unwind when the cell is exposed to fluid flow (Rangel et al., 2013).

Correspondingly, in this study we find that the unwinding force is significantly different for the WT and the A22R mutant and this mutant is severely attenuated in bladder and gut colonization.

2) When correlating the in vitro and in vivo results, there are several uncontrolled variables that need to be addressed including proteases, bacterial clumping, number of pili versus structure of pili for fimA mutants.

The reviewers are correct in pointing out that there are uncontrolled variables in our in vitro and in vivo experiments. in vitro bacterial clumping and pilus expression are variable between FimA mutants, however, we show via hemagglutination assays that this does not impact the function of these pili in vitro. We do find differences between WT and mutant FimA expressing strains in both two different niches within the urinary tract (the bladder and the kidney) and within the gut. In allin vivo mouse experiments, there are numerous uncontrolled variables present, including host and bacterial proteases. While we do not control for these proteins in our studies, we note that we use the same strain of mice for each experiment and biological replicates for experiments are performed in mice that are the same age, from the same vendor and batch, and have been exposed to the same conditions and diet in our vivarium. While host proteases might alter pathogenesis, our WT and mutant mice should be exposed to the same protease conditions. We have now added the possibility to the Discussion that the uncoiling of the A22R mutant might make sites accessible to host proteases that would not be available in the coiled wt rod. As we note, this does not change the conclusion of our paper that the mechanical properties of the rod have been “tuned” over evolution as an adaptation to specific environments. Further, while we feel that the appearance of phenotypes such as clumping or alterations in pilus expression highlights the role of residues in interactions within type 1 pili, the in vitro measured phenotype that correlates best with in vivo virulence is the optical tweezer experiments. Note, that we have now removed results of the mutant E121R, which had the most clumping phenotype (see below).

Further detail can be found in the separate reviews below. If you have any questions about the revisions, please let us know.Reviewer #1:The only concern of this reviewer is that some leaps are made to correlate observations in vitro, with those in vivo. The suggestion that the spring-like qualities of the pilus rod are critical for pathogenesis is entirely possible, but is far from proven.

Please see response to essential revision 1.

Moreover, few specifics are provided that relate the actual shear forces of urine at various distances from the mucosal boundary to the forces measured for variations in the protein.

Urine flow in the urinary tract is indeed complex and hash: boluses are transported from the kidney to the bladder via peristaltic motion; and high fluid flow exposes bacterial cells to high shear forces during expulsion of urine from the bladder. In addition, peristaltic motion creates shear forces that include reversal wall shear forces and normal forces. Bacteria need to attach and adhere in these conditions, which implies that they are not more than a “coiled-pili-length” away from the surface. Therefore, the shear forces, which increase more-or-less linearly with height for small displacement, will not be significantly different at various distances from the mucosal boundary. This implies also for flow chamber experiments.

Several flow chamber experiments generating shear stress of similar magnitude as physiological conditions (Busscher and van der Mei, 2006) have been performed (Aprikian et al., 2011; Rangel et al., 2013)). Interestingly, these experiments exposed cells from low to high forces, that is, from 10 – 140 pN (Aprikian et al., 2011), which is in the range of WT type 1 pili uncoiling. Pili uncoiling for WT bacteria, leading to bacterial displacement of the order of a few micrometers, have also been demonstrated in flow chamber experiments. Thus experiments support that shear forces exposing bacteria within physiological conditions are sufficiently to uncoil WT pili (Rangel et al., 2013). Since we measured lower uncoiling force for the mutants, we expect that these “softer-spring” pili attaching bacteria to cells will not be able to modulate and dampen the fluid forces as good as WT pili. This reduced damping capability will therefore make it harder for bacteria to attach and adhere.

A number of uncontrolled variables could explain why mutant variants of the protein are less effective in vivo, including increased susceptibility to tissue resident proteases, altered interactions with various elements of the mucosa, altered physical states imposed (such as the clumping that was observed), etc. While an interesting prospect, the mechanical properties proposed could be explored further for proof, or the discussion of this prospect modified in the text (although the latter arguably could affect the perceived impact of the study in altering existing paradigms).

Please see response to essential revisions 1 and 2.

Aside from that concern, this reviewer found the paper to be of considerable interest and well done.Reviewer #2:[…] 1) The A22R mutant has a significant defect at 6 hours in the mouse. Does this mutant affect the ability of bacteria to invade the superficial facet cells?

This is an important point. To address this question we now include data regarding the ability of the A22R mutant to invade into bladder tissue at 1, 3, and 6 hours. This was done via gentamycin protection assays. We found that the A22R mutant does indeed have an invasion defect compared to WT starting at 1 hpi. This ability of this strain to persist in the lumen of the bladder is also attenuated compared to WT. We have added this invasion data to Figure 5 as panel G.

2) A few of the mutants were tested in vivo, but not in vitro. For example, E123R was tested in the mouse and displayed a defect in acute and chronic UTI (and a slight defect in gut colonization), but the force measurements for unwinding this pili were not determined.

Thank you for pointing this out. The E123R mutant was not included in the optical tweezer data due to difficulty in obtaining reading with single pili attached the beads. The attachment of a single pilus to a bead is essential for us to calculate an accurate unwinding force measurement. After multiple additional failed attempts to acquire this data for this FimA mutant, we have decided to remove all analysis of this mutant from the paper.

Reviewer #3:[…] 1) Measurement of the unwinding force.a) The authors state in the Materials and methods part they expect "attachment a few pili with the microsphere". Determination of the unwinding force, however, is only possible it the authors can make sure that only a single pilus is attached at the time.

We thank the reviewer for bringing this issue to our attention. Based on this comment, it is clear that our description of the molecular tweezers experiment in the Materials and methods section did not clearly explain how the force-spectroscopy measurements were taken. Therefore, we have now included additional text in the Materials and methods to clarify the experimental set up for this method. Further, we have included an additional figure (Figure 4—figure supplement 2) that better explains how we assessed the data when multi-pili interactions to the bead occur. Further explanation of multi-pili attachment to microspheres and analysis of force curves can be found in references (Andersson et al., 2008; Castelain et al.; Fallman et al., 2005; Jass et al., 2004; Mortezaei et al., 2015).

The new text reads:

“A free-floating bacterium was trapped by the optical tweezers run at low power to avoid cell damage. […] Finally, we controlled the piezo-stage and sampled the data using an in-house LabView program (Andersson, 2017).”

b) I haven't found any information about how the mutations analysed in Figure 4 affect the number of pili per cell. If more than a pilus is bound during a force measurement, then the level of piliation can affect the measured uncoiling force.

Thank you for this comment, we have provided additional text in the Materials and methods section that explicitly states our use of only a single pilus to make any force measurements. Please see response above to point 1a.

c) Only technical replicates from a single sample are inacceptable. Data should be acquired on biological replicates on at least three different days.

This is an excellent and critical point made by the reviewer. We have included additional data into Figure 4 to ensure that biological replicates from different days are shown.

2) Expression levels and functionality of pili with point mutations.The authors state that "FimA mutant strain produced minimal changes to the levels of type 1 piliation […] and to the function of the type 1 pilus". The experimental data supporting this statement is shown in Figure 5—figure supplement 1. I cannot understand this statement based on the data. The percentage of "bald" bacteria is about 3x higher for the D116R strain compared to wt. The mannose-inhibitable HA-titre is about 10x higher in the E123R and D116R strains compared to wt. We can expect that the pilus density affects host-pathogen interactions. One way of showing that the 3x increase of pilus density does not affect the conclusion, would be to generate strains with wt fimA sequence but decreased pilus levels.

In order to better characterize the variability that we observe in pilus expression with each mutant, we have altered the text in the manuscript to more explicitly reflect the data presented in Figure 5—figure supplement 1. However, we feel that our overall statement that indicated FimA mutants do not prevent the expression or function of type 1 pili in vitrois demonstrated.

This text now reads:

“When compared to a strain with the reintegrated WT UTI89 fimA sequence, we found that all reintegrated FimA mutant strains demonstrated similar levels of GP-RBC agglutination, in vitro (Figure 5—figure supplement 1),despite some variability in the level of piliation between FimA mutant strain (Figure 5—figure supplement 1).[…] Overall, these findings indicate that the identified FimA mutations do not prevent the expression or function of type 1 pili in vitro.”

3) Comparison between different strains.a) The authors explain that they used different bacterial strains with different fimA sequences. This is probably fine, but the way the data is presented now, I get confused with the different notations of the point mutations. For example, is the A22R mutation in Figure 4 comparable to A24R in Figure 5? Same for D114R and D116R etc.? Can the authors come up with a smart way of indicating these connections within the figures?

This is an excellent point that was made by multiple reviewers. In order to prevent confusion, we have chosen to refer to all residues from both strains using the BW25113 number system.

We have also addressed this change in the text:

“Thus, we do note that FimA_UTI89_ differs from our cryo-EM rod FimA_BW25113_ sequence in 16 residues, with all of the differences lying within highly variable residues located on the exterior of the pilus rod, including two additional amino acids at the N-terminus. […] To avoid confusion herein all residues are discussed using the BW25113 numbering system.”